# DiCo: Revitalizing ConvNets for Scalable and Efficient Diffusion Modeling

**Yuang Ai**[1,2]  **Qihang Fan**[1,2]  **Xuefeng Hu**[3]  **Zhenheng Yang**[3]  **Ran He**[1,2]  **Huaibo Huang**[1,2*]

[1]CASIA    [2]UCAS    [3]ByteDance

Code and models: https://github.com/shallowdream204/DiCo

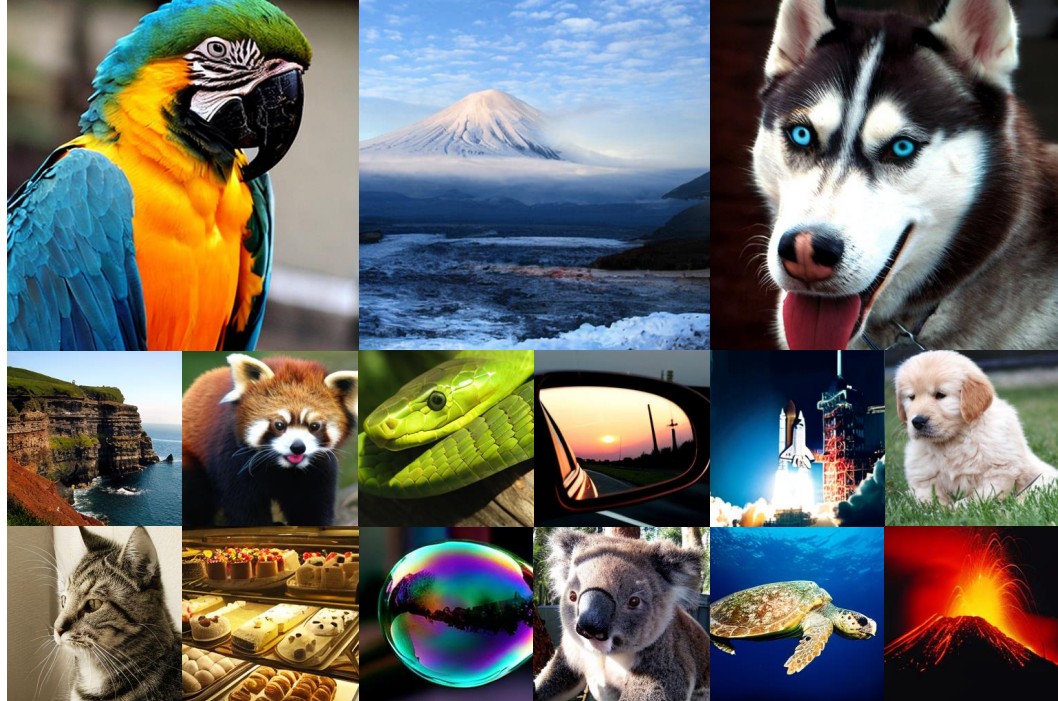

Figure 1: ***Diffusion ConvNet* achieves superior image quality with high efficiency.** We show samples from two of our DiCo-XL models trained on ImageNet at $512{\times}512$ and $256{\times}256$ resolution.

## Abstract

Diffusion Transformer (DiT), a promising diffusion model for visual generation, demonstrates impressive performance but incurs significant computational overhead. Intriguingly, analysis of pre-trained DiT models reveals that global self-attention is often redundant, predominantly capturing local patterns—highlighting the potential for more efficient alternatives. In this paper, we revisit convolution as an alternative building block for constructing efficient and expressive diffusion models. However, naively replacing self-attention with convolution typically results in degraded performance. Our investigations attribute this performance gap to the higher channel redundancy in ConvNets compared to Transformers. To resolve this, we introduce a compact channel attention mechanism that promotes the activation of more diverse channels, thereby enhancing feature diversity. This leads to Diffusion ConvNet (DiCo), a family of diffusion models built entirely from

*Corresponding author: Huaibo Huang <huaibo.huang@cripac.ia.ac.cn>

39th Conference on Neural Information Processing Systems (NeurIPS 2025).

standard ConvNet modules, offering strong generative performance with significant efficiency gains. On class-conditional ImageNet generation benchmarks, DiCo-XL achieves an FID of 2.05 at 256×256 resolution and 2.53 at 512×512, with a **2.7×** and **3.1×** speedup over DiT-XL/2, respectively. Furthermore, experimental results on MS-COCO demonstrate that the purely convolutional DiCo exhibits strong potential for text-to-image generation.

# 1  Introduction

Diffusion models [73, 75, 33, 74, 76] have sparked a transformative advancement in generative learning, demonstrating remarkable capabilities in synthesizing highly photorealistic visual content. Their versatility and effectiveness have led to widespread adoption across a broad spectrum of real-world applications, including text-to-image generation [66, 69, 67], image editing [59, 46, 10], image restoration [45, 3, 4], video generation [36, 88, 7], and 3D content creation [64, 87, 84].

Early diffusion models (e.g., ADM [14] and Stable Diffusion [67]) primarily employed hybrid U-Net [68] architectures that integrate convolutional layers with self-attention. More recently, Transformers [83] have emerged as a more powerful and scalable backbone [62, 6], prompting a shift toward fully Transformer-based designs. As a result, Diffusion Transformers (DiTs) are gradually supplanting traditional U-Nets, as seen in leading diffusion models such as Stable Diffusion 3 [20], FLUX [49], and Sora [9]. However, the quadratic computational complexity of self-attention presents substantial challenges, especially for high-resolution image synthesis. Recent efforts [100, 79, 25, 63, 91, 2] have explored more efficient alternatives, focusing on linear-complexity RNN-like architectures, such as Mamba [26] and Gated Linear Attention [92]. While these models improve efficiency, their causal design inherently conflicts with the bidirectional nature of visual generation [30, 55], limiting their effectiveness. Furthermore, as illustrated in Fig. 3, even with highly optimized CUDA implementations, their runtime advantage over conventional DiTs remains modest in high-resolution settings. This leads us to a key question: *Is it possible to design a hardware-efficient diffusion backbone that also preserves strong generative capabilities like DiTs?*

To approach this question, we begin by examining the characteristics that underlie the generative power of DiTs. In visual recognition tasks, the success of Vision Transformers [18] is often credited to the self-attention's ability to capture long-range dependencies [42, 23, 22]. However, in generative tasks, we observe a different dynamic. As depicted in Fig. 4, for both pre-trained class-conditional (DiT-XL/2 [62]) and text-to-image (PixArt-$\alpha$ [12] and FLUX [49]) DiT models, when queried with an anchor token, attention predominantly concentrates on nearby spatial tokens, largely disregarding distant ones. This finding suggests that computing global attention may be redundant for generation, underscoring the significance of local spatial modeling. Unlike recognition tasks, where long-range interactions are critical for global semantic reasoning, generative tasks appear to emphasize fine-grained texture and local structural fidelity. These observations reveal the inherently localized nature of attention in DiTs and motivate the pursuit of more efficient architectures.

In this work, we revisit convolutional neural networks (ConvNets) and propose Diffusion ConvNet (DiCo), a simple yet highly efficient convolutional backbone tailored for diffusion models. Compared to self-attention, convolutional operations are more hardware-friendly, offering significant advantages for large-scale and resource-constrained deployment. While substituting self-attention with convolution substantially improves efficiency, it typically results in degraded performance. As illustrated in Fig. 5, this naive replacement introduces pronounced channel redundancy, with many channels remaining inactive during generation. We hypothesize that this stems from the inherently stronger representational capacity of self-attention compared to convolution. To address this, we introduce a compact channel attention (CCA) mechanism, which dynamically activates informative channels with lightweight linear projections. As a channel-wise global modeling approach, CCA enhances the model's representational capacity and feature diversity while maintaining low computational overhead. Unlike modern recognition ConvNets that rely on large, costly kernels [15, 28], DiCo adopts a streamlined design based entirely on efficient 1×1 pointwise convolutions and 3×3 depthwise convolutions. Despite its architectural simplicity, DiCo delivers strong generative performance.

As shown in Fig. 2 and Fig. 3, DiCo models outperform recent diffusion models on both the ImageNet 256×256 and 512×512 benchmarks. Notably, our DiCo-XL models achieve impressive FID scores of 2.05 and 2.53 at 256×256 and 512×512 resolution, respectively. In addition to performance gains,

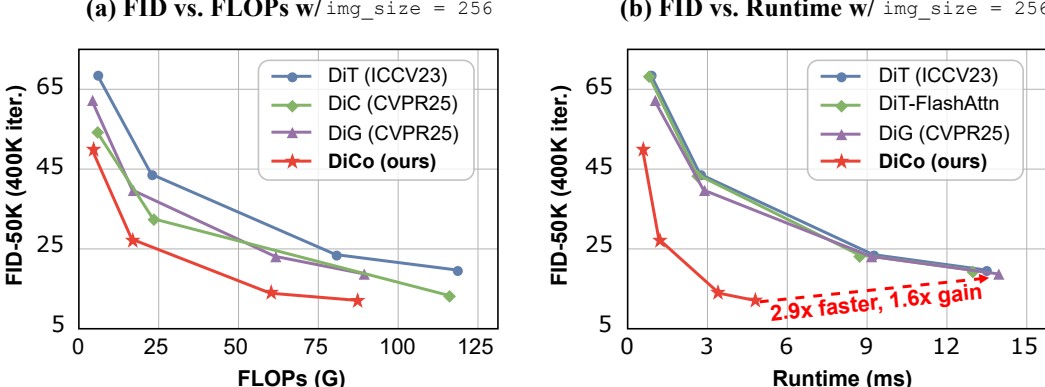

Figure 2: **Comparison of performance and efficiency** with recent diffusion models (DiT [62], DiC [81], and DiG [100]) on ImageNet 256×256. Our proposed DiCo achieves the best performance while maintaining high efficiency. Compared to DiG-XL/2 with CUDA-optimized Flash Linear Attention [92], DiCo-XL runs 2.9× faster and achieves a 1.6× improvement in FID.

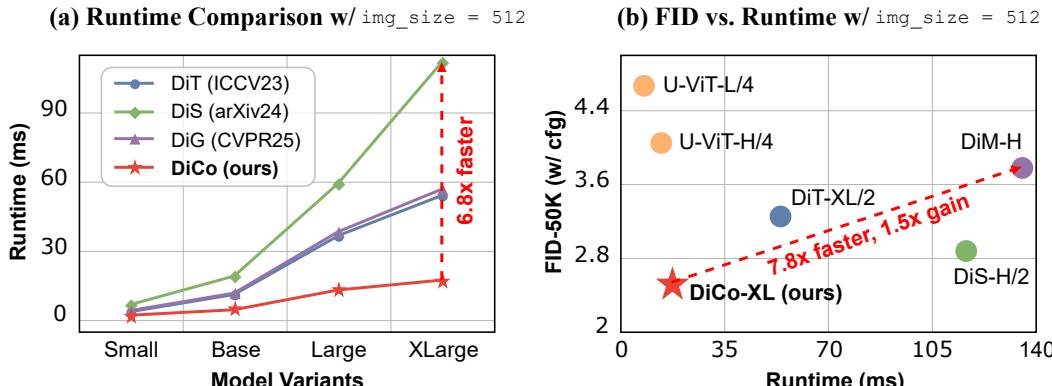

Figure 3: **(a) Runtime comparison** between DiT [62], DiS [25] (with Mamba [26]), DiG [100] (with Gated Linear Attention [92]), and our DiCo at 512×512 resolution. DiCo is 3.3× faster than DiS at the small model scale and 6.8× faster at the XL scale. **(b) FID vs. runtime** of various methods on ImageNet 512×512. DiCo-XL achieves an FID of 2.53 while maintaining high efficiency.

DiCo models exhibit considerable efficiency advantages over attention-based [83], Mamba-based [26], and linear attention-based [44] diffusion models. Specifically, at 256×256 resolution, DiCo-XL achieves a 26.4% reduction in Gflops and is 2.7× faster than DiT-XL/2 [62]. At 512×512 resolution, DiCo-XL operates 7.8× and 6.7× faster than the Mamba-based DiM-H [79] and DiS-H/2 [25] models, respectively. Our largest model, DiCo-H with 1 billion parameters, further reduces the FID on ImageNet 256×256 to 1.90. In addition, we validate the applicability of DiCo for text-to-image generation on the MS-COCO dataset. These results collectively highlight the strong potential of DiCo in diffusion-based generative modeling.

Overall, the main contributions of this work can be summarized as follows:

- We analyze pre-trained DiT models and reveal significant redundancy and locality within their global attention mechanisms. These findings may inspire researchers to develop more efficient strategies for constructing high-performing diffusion models.

- We propose DiCo, a simple, efficient, and powerful ConvNet backbone for diffusion models. By incorporating compact channel attention, DiCo significantly improves representational capacity and feature diversity without sacrificing efficiency.

- We conduct extensive experiments on class-conditional ImageNet benchmarks. DiCo outperforms recent diffusion models in both generation quality and speed. Furthermore, the purely convolutional DiCo demonstrates strong potential in text-to-image generation.

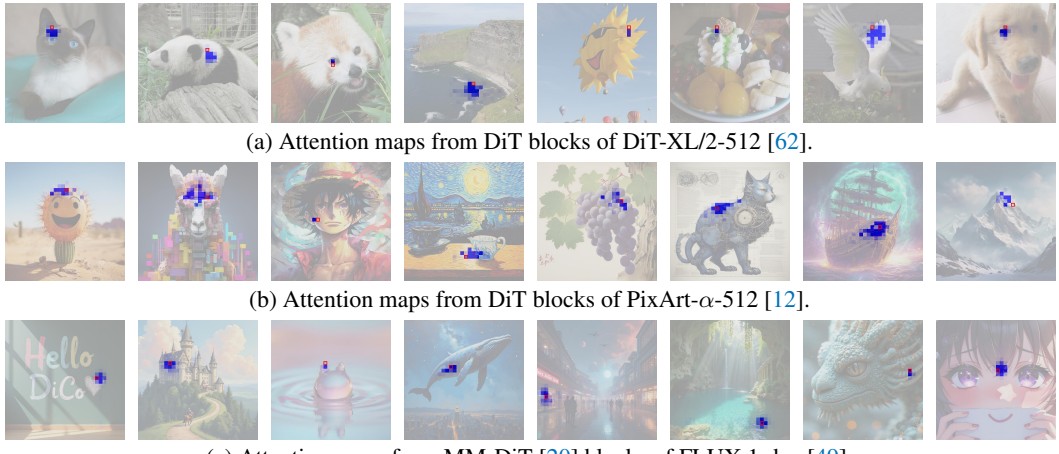

(a) Attention maps from DiT blocks of DiT-XL/2-512 [62].

(b) Attention maps from DiT blocks of PixArt-α-512 [12].

(c) Attention maps from MM-DiT [20] blocks of FLUX.1-dev [49].

Figure 4: **Visualization of attention maps from well-known DiT models**. The intensity of the blue color indicates the magnitude of attention scores. For self-attention across different layers in these models, only a few neighboring tokens contribute significantly to the attention distribution of a given anchor token (red box), resulting in highly redundant and localized representations.

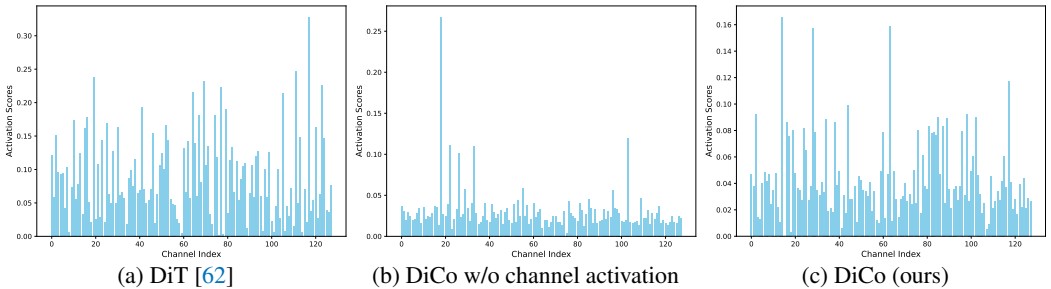

(a) DiT [62]   (b) DiCo w/o channel activation   (c) DiCo (ours)

Figure 5: **Comparison of channel activation scores across different diffusion models.** Channel activation scores are computed using ReLU followed by global average pooling on the final layer's self-attention or convolution outputs [102]. Directly replacing self-attention in DiT with convolution introduces significant channel redundancy, as most channel activation scores remain at low levels.

## 2   Related Work

**Architecture of Diffusion Models.** Early diffusion models commonly employ U-Net [68] as the foundational architecture [14, 34, 67]. More recently, a growing body of research has explored Vision Transformers (ViTs) [18] as alternative backbones for diffusion models, yielding remarkable results [62, 6, 58, 96, 65, 54]. Notably, DiT [62] has demonstrated the excellent performance of transformer-based architectures, achieving SOTA performance on ImageNet generation. However, the quadratic computational complexity inherent in ViTs presents significant challenges in terms of efficiency for long sequence modeling. To mitigate this, recent studies have explored the use of RNN-like architectures with linear complexity, such as Mamba [26] and linear attention [44], as backbones for diffusion models [25, 100, 79, 91, 63]. DiS [25] and DiM [79] employ Mamba to reduce computational overhead, while DiG [100] leverages Gated Linear Attention [92] to achieve competitive performance with improved efficiency. In this work, we revisit ConvNets as backbones for diffusion models. We show that, with proper design, pure convolutional architectures can achieve superior generative performance, providing an efficient and powerful alternative to DiTs.

**ConvNet Designs.** Over the past decade, convolutional neural networks (ConvNets) have achieved remarkable success in computer vision [31, 41, 89, 5, 19]. Numerous lightweight ConvNets have been developed for real-world deployment [39, 71, 38, 16]. Although Transformers have gradually become the dominant architecture across a wide range of tasks, their substantial computational overhead remains a significant challenge. Many modern ConvNet designs achieve competitive performance

while maintaining high efficiency. ConvNeXt [57] explores the modernization of standard ConvNets and achieves superior results compared to transformer-based models. RepLKNet [15] investigates the use of large-kernel convolutions, expanding kernel sizes up to $31 \times 31$. UniRepLKNet [17] further generalizes large-kernel ConvNets to domains such as audio, point clouds, and time-series forecasting. In this work, we explore the potential of pure ConvNets for diffusion-based image generation, and show that simple, efficient ConvNet designs can also deliver excellent performance.

# 3 Method

## 3.1 Preliminaries

**Diffusion formulation.** We first revisit essential concepts underpinning diffusion models [33, 76]. Diffusion models are characterized by a forward noising procedure that progressively injects noise into a data sample $x_0$. Specifically, this forward process can be expressed as:

$$q(x_{1:T}|x_0) = \prod_{t=1}^{T} q(x_t|x_{t-1}), q(x_t|x_0) = \mathcal{N}(x_t; \sqrt{\bar{\alpha}_t}x_0, (1-\bar{\alpha}_t)\mathbf{I}), \tag{1}$$

where $\bar{\alpha}_t$ are predefined hyperparameters. The objective of a diffusion model is to learn the reverse process: $p_\theta(x_{t-1}|x_t) = \mathcal{N}(\mu_\theta(x_t), \Sigma_\theta(x_t))$, where neural networks parameterize the mean and covariance of the process. The training involves optimizing a variational lower bound on the log-likelihood of $x_0$, which simplifies to:

$$\mathcal{L}(\theta) = -p(x_0|x_1) + \sum_t \mathcal{D}_{KL}(q^*(x_{t-1}|x_t, x_0)||p_\theta(x_{t-1}|x_t)). \tag{2}$$

To simplify training, the model's predicted mean $\mu_\theta$ can be reparameterized as a noise predictor $\epsilon_\theta$. The objective then reduces to a straightforward mean-squared error between the predicted noise and the true noise $\epsilon_t$: $\mathcal{L}_{simple}(\theta) = ||\epsilon_\theta(x_t) - \epsilon_t||_2^2$. Following DiT [62], we train the noise predictor $\epsilon_\theta$ using the simplified loss $\mathcal{L}_{simple}$, while the covariance $\Sigma_\theta$ is optimized using the full loss $\mathcal{L}$.

**Classifier-free guidance.** Classifier-free guidance (CFG) [35] is an effective method to enhance sample quality in conditional diffusion models. It achieves such enhancement by guiding the sampling process toward outputs strongly associated with a given condition $c$. Specifically, it modifies the predicted noise to obtain high $p(x|c)$ as: $\hat{\epsilon}_\theta(x_t, c) = \epsilon_\theta(x_t, \emptyset) + s \cdot \nabla_x \log p(x|c) \propto \epsilon_\theta(x_t, \emptyset) + s \cdot (\epsilon_\theta(x_t, c) - \epsilon_\theta(x_t, \emptyset))$. where $s \geq 1$ controls the guidance strength, and $\epsilon_\theta(x_t, \emptyset)$ is an unconditional prediction obtained by randomly omitting the conditioning information during training. Following prior works [62, 100], we adopt this technique to enhance the quality of generated samples.

## 3.2 Network Architecture

Currently, diffusion models are primarily categorized into three architectural types: (1) Isotropic architectures without any downsampling layers, as seen in DiT [62]; (2) Isotropic architectures with long skip connections, exemplified by U-ViT [6]; and (3) U-shaped architectures, such as U-DiT [82]. Motivated by the crucial role of multi-scale features in image denoising [97, 1], we adopt a U-shaped design to construct a hierarchical model. We also conduct an extensive ablation study to systematically compare the performance of these different architectural choices in Table 4.

As illustrated in Fig. 6 (a), DiCo employs a three-stage U-shaped architecture composed of stacked DiCo blocks. The model takes the spatial representation $z$ generated by the VAE encoder as input. For an image of size $256 \times 256 \times 3$, the corresponding $z$ has dimensions $32 \times 32 \times 4$. To process this input, DiCo applies a $3 \times 3$ convolution that transforms $z$ into an initial feature map $z_0$ with $D$ channels. For conditional information—specifically, the timestep $t$ and class label $y$—we employ a multi-layer perceptron (MLP) and an embedding layer, serving as the timestep and label embedders, respectively. At each block $l$ within DiCo, the feature map $z_{l-1}$ is passed through the $l$-th DiCo block to produce the output $z_l$.

Within each stage, skip connections between the encoder and decoder facilitate efficient information flow across intermediate features. After concatenation, a $1 \times 1$ convolution is applied to reduce the channel dimensionality. To enable multi-scale processing across stages, we utilize pixel-unshuffle operations for downsampling and pixel-shuffle operations for upsampling. Finally, the output feature $z_L$ is normalized and passed through a $3 \times 3$ convolutional head to predict both noise and covariance.

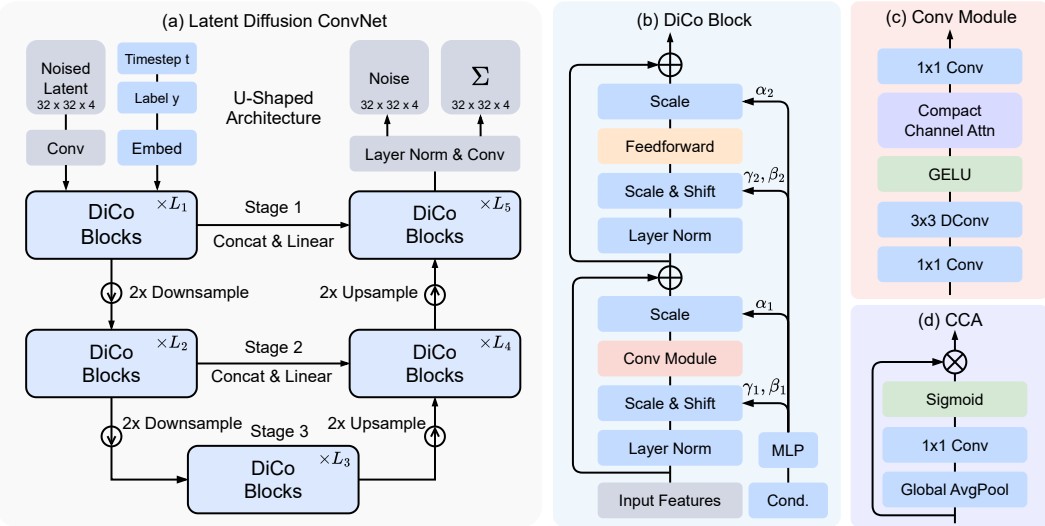

Figure 6: **Architecture of DiCo,** which consists of (b) DiCo Block, (c) Conv Module, and (d) Compact Channel Attention (CCA). DConv denotes depthwise convolution.

### 3.3 DiCo Block

**Motivation.** As shown in Fig. 4, the self-attention computation in DiT models—whether for class-conditional or text-to-image generation—exhibits a distinctly local structure and significant redundancy. This observation motivates us to replace the global self-attention in DiT with more hardware-efficient operations. A natural alternative is convolution, which is well-known for its ability to efficiently model local patterns. We first attempt to substitute self-attention with a combination of $1 \times 1$ pointwise convolutions and $3 \times 3$ depthwise convolutions.

However, the direct replacement leads to a degradation in generation performance. As shown in Fig. 5, compared to DiT, many channels in the modified model remain inactive, indicating substantial channel redundancy. We hypothesize that this performance drop stems from the fact that self-attention, being dynamic and content-dependent, provides greater representational power than convolution, which relies on static weights. To address this limitation, we introduce a compact channel attention mechanism to dynamically activate informative channels. We describe the full design in detail below.

**Block designs.** The core design of DiCo is centered around the Conv Module, as shown in Fig. 6 (c). We first apply a $1 \times 1$ convolution to aggregate pixel-wise cross-channel information, followed by a $3 \times 3$ depthwise convolution to capture channel-wise spatial context. A GELU activation is employed for non-linear transformation. To further address channel redundancy, we introduce a compact channel attention (CCA) mechanism to activate more informative channels. As illustrated in Fig. 6 (d), CCA first aggregates features via global average pooling (GAP) across the spatial dimensions, then applies a learnable $1 \times 1$ convolution followed by a sigmoid activation to generate channel-wise attention weights. Generally, the whole process of Conv Module can be described as:

$$Y = W_{p_2}\text{CCA}(\text{GELU}(W_d W_{p_1} X)), \text{CCA}(X) = X \odot \text{Sigmoid}(W_p\text{GAP}(X)), \quad (3)$$

where $W_{p_{(\cdot)}}$ is the $1 \times 1$ point-wise conv, $W_d$ is the depthwise conv, and $\odot$ denotes the channel-wise multiplication. As shown in Fig. 5 (c), this simple and efficient design effectively reduces feature redundancy and enhances the representational capacity of the model. To incorporate conditional information from the timestep and label, we follow DiT by adding the input timestep embedding $t$ and label embedding $y$, and using them to predict the scale parameters $\alpha, \gamma$ and the shift parameter $\beta$.

**Modification for text-to-image.** We investigate two different approaches for incorporating textual features into DiCo. The first uses the widely adopted cross-attention [12] mechanism, integrated into the DiCo architecture to fuse text and visual features. The second transforms CLIP text embeddings into dynamic depthwise convolution (DWC) kernels. We pad the text embeddings to a length of 81, feed them through a learnable MLP, and reshape the output into a $9 \times 9$ kernel. This kernel dynamically modulates DiCo's features via depthwise convolution. In this way, we can construct a

**Algorithm 1** PyTorch code of text conditional depthwise convolution

```python
import torch
import torch.nn.functional as F

def text_conditional_dwconv(x, context):
    # x: (B, C, H, W) input feature maps
    # context: (B, 77, C) CLIP text embeddings after an MLP
    # output: (B, C, H, W) output after depthwise convolution
    B, C, H, W = x.shape
    context_pad = torch.cat([context, context[:, -1:].expand(-1, 4, -1)], dim=1) # (B, 81, C)
    kernels = context_pad.reshape(B, 9, 9, C).permute(0, 3, 1, 2).reshape(B * C, 1, 9, 9)
    x_flat = x.view(1, B * C, H, W)
    output = F.conv2d(x_flat, kernels, padding=4, groups=B * C).view(B, C, H, W)
    return output
```

fully convolutional text-to-image DiCo model without relying on any self-attention or cross-attention operations. We provide its detailed PyTorch implementation in Algorithm 1. Both feature fusion modules are inserted after the Conv Module within each DiCo block.

### 3.4 Architecture Variants

We establish four model variants—DiCo-S, DiCo-B, DiCo-L, and DiCo-XL—whose parameter counts are aligned with those of DiT-S/2, DiT-B/2, DiT-L/2, and DiT-XL/2, respectively. Compared to their DiT counterparts, our DiCo models achieve a significant reduction in computational cost, with Gflops ranging from only 70.1% to 74.6% of those of DiT. Furthermore, to explore the potential of our design, we scale up DiCo to 1 billion parameters, resulting in DiCo-H. The architectural configurations of these models are detailed in Appendix Table 6.

## 4 Experiments

### 4.1 Experimental Setup

**Datasets and Metrics.** Following previous works [62, 100, 81], we conduct experiments on class-conditional ImageNet-1K [13] generation benchmark at 256×256 and 512×512 resolutions. We use the Fréchet Inception Distance (FID) [32] as the primary metric to evaluate model performance. In addition, we report the Inception Score (IS) [70], Precision, and Recall [48] as secondary metrics. All these metrics are computed using OpenAI's TensorFlow evaluation toolkit [14].

**Implementation Details.** For DiCo-S/B/L/XL, *we adopt exactly the same experimental settings as used for DiT.* Specifically, we employ a constant learning rate of $1 \times 10^{-4}$, no weight decay, and a batch size of 256. The only data augmentation applied is random horizontal flipping. We maintain an exponential moving average (EMA) of the DiCo weights during training, with a decay rate of 0.9999. The pre-trained VAE [67] is used to extract latent features. For our largest model, DiCo-H, we follow the training settings of U-ViT [6], increasing the learning rate to $2 \times 10^{-4}$ and scaling the batch size to 1024 to accelerate training. Additional details are provided in Appendix Sec. B.

### 4.2 Main Results

**Comparison under the DiT Setting.** In addition to DiT [62], we also select recent diffusion models, DiG [100] and DiC [81], as baselines, since they similarly follow the experimental setup of DiT.Table 1 presents the comparison results on ImageNet 256×256. Across different model scales trained for 400K iterations, our DiCo consistently achieves the best or second-best performance across all metrics. Furthermore, when using classifier-free guidance (CFG), our DiCo-XL achieves an FID of 2.05 and an IS of 282.17. Beyond performance improvements, DiCo also demonstrates significant efficiency gains compared to both the baselines and Mamba-based models.

Table 2 presents the results on ImageNet 512×512. At higher resolutions, our model demonstrates greater improvements in both performance and efficiency. Specifically, DiCo-XL achieves an FID of 2.53 and an IS of 275.74, while reducing Gflops by 33.3% and achieving a 3.1× speedup compared to DiT-XL/2. These results highlight that our convolutional architecture remains highly efficient and effective for high-resolution image generation.

Table 1: **Comparison under the DiT setting on ImageNet 256×256.** The performance at 400K training steps is reported without CFG for early-stage comparison. We mark the best results for each model scale in bold. Throughput (image/s) is measured on A100 with batch size 64 at fp32 precision. DiT and DiG are optimized with FlashAttention-2 and Flash Linear Attention, respectively.

| Model | Token Mixing Type | Gflops | Throughput | FID↓ | IS↑ | Pre.↑ | Rec.↑ |
|---|---|---|---|---|---|---|---|
| ADM-U [14] | Conv + Attn | 742 | - | 3.94 | 215.84 | 0.83 | 0.53 |
| LDM-4 [67] | Conv + Attn | - | - | 3.95 | 178.22 | 0.81 | 0.55 |
| U-ViT-H/2 [6] | Attn | 133.25 | 73.45 | 2.29 | 263.88 | 0.82 | 0.57 |
| *Mamba-based diffusion models.* | | | | | | | |
| DiM-H [79] | Conv + SSM | 210 | 25.06 | 2.21 | - | - | - |
| DiS-H/2 [25] | Conv + SSM | - | 33.95 | 2.10 | 271.32 | 0.82 | 0.58 |
| DiffuSSM-XL [91] | SSM | 280.3 | - | 2.28 | 259.13 | 0.86 | 0.56 |
| DiMSUM-L/2 [63] | Attn + SSM | 84.49 | 59.13 | 2.11 | - | - | 0.59 |
| *Baselines and Ours (w/ the same hyperparameters).* | | | | | | | |
| DiT-S/2 (400K) [62] | Attn | 6.06 | 1234.01 | 68.40 | - | - | - |
| DiC-S (400K) [81] | Conv | 5.9 | - | 58.68 | 25.82 | - | - |
| DiG-S/2 (400K) [100] | Conv + Attn | 4.30 | 961.24 | 62.06 | 22.81 | 0.39 | 0.56 |
| **DiCo-S** (400K) | Conv | **4.25** | **1695.73** | **49.97** | **31.38** | **0.48** | **0.58** |
| DiT-B/2 (400K) | Attn | 23.02 | 380.11 | 43.47 | - | - | - |
| DiC-B (400K) | Conv | 23.5 | - | 32.33 | 48.72 | - | - |
| DiG-B/2 (400K) | Conv + Attn | 17.07 | 345.89 | 39.50 | 37.21 | 0.51 | **0.63** |
| **DiCo-B** (400K) | Conv | **16.88** | **822.97** | **27.20** | **56.52** | **0.60** | 0.61 |
| DiT-L/2 (400K) | Attn | 80.73 | 114.63 | 23.33 | - | - | - |
| DiG-L/2 (400K) | Conv + Attn | 61.66 | 109.01 | 22.90 | 59.87 | 0.60 | **0.64** |
| **DiCo-L** (400K) | Conv | **60.24** | **288.32** | **13.66** | **91.37** | **0.69** | 0.61 |
| DiT-XL/2 (400K) | Attn | 118.66 | 76.90 | 19.47 | - | - | - |
| DiC-XL (400K) | Conv | 116.1 | - | 13.11 | 100.15 | - | - |
| DiG-XL/2 (400K) | Conv + Attn | 89.40 | 71.74 | 18.53 | 68.53 | 0.63 | **0.64** |
| **DiCo-XL** (400K) | Conv | **87.30** | **208.47** | **11.67** | **100.42** | **0.71** | 0.61 |
| DiT-XL/2 (w/ CFG) | Attn | 118.66 | 76.90 | 2.27 | 278.24 | 0.83 | 0.57 |
| DiG-XL/2 (w/ CFG) | Conv + Attn | 89.40 | 71.74 | 2.07 | 278.95 | 0.82 | **0.60** |
| **DiCo-XL** (w/ CFG) | Conv | **87.30** | **208.47** | **2.05** | **282.17** | **0.83** | 0.59 |
| DiC-H (w/ CFG) | Conv | 204.4 | - | 2.25 | - | - | - |
| **DiCo-H** (w/ CFG) | Conv | **194.15** | **117.57** | **1.90** | **284.31** | **0.83** | **0.61** |

**Scaling the model up.** To further explore the potential of our model, we scale it up to 1 billion parameters. As shown in Table 1, compared to DiCo-XL, the larger DiCo-H achieves further improvements in FID (1.90 vs. 2.05), demonstrating the great scalability of our architecture. More comparison results and generated images can be found in the Appendix Sec. D and Sec. E.

**Text-to-Image Generation.** We follow [6] to conduct small-scale text-to-image generation experiments. Specifically, we adopt the same experimental setup as [96]: training and evaluating models from scratch on MS-COCO [53], using CLIP as the text encoder with a token length of 77.

As shown in Table 3, our DiCo achieves superior generation quality for text-to-image generation. Notably, using text conditional DWC in place of cross-attention further improves throughput while maintaining competitive performance. This suggests that the fully convolutional DiCo has the potential to serve as the backbone for large-scale text-to-image diffusion models.

## 4.3 Ablation Study

For the ablation study, we use the small-scale model and evaluate performance on the ImageNet 256×256 benchmark to enable fast training speed. All models are trained for 400K iterations and evaluated without CFG. Notably, in this section, self-attention in DiT is not accelerated using FlashAttention-2 to ensure a fair speed comparison with other efficient attention mechanisms.

Table 2: **Comparison under the DiT setting on ImageNet 512×512.** We mark the best performance with CFG in bold. The performance at 1.3M/3M training steps is reported without using CFG.

| Model | Token Mixing Type | Gflops | Throughput | FID↓ | IS↑ | Pre.↑ | Rec.↑ |
|---|---|---|---|---|---|---|---|
| ADM-U [14] | Conv + Attn | 2813 | - | 3.85 | 221.72 | 0.84 | 0.53 |
| U-ViT-L/4 [6] | Attn | **76.52** | **128.49** | 4.67 | 213.28 | **0.87** | 0.45 |
| U-ViT-H/4 [6] | Attn | 133.27 | 73.42 | 4.05 | 263.79 | 0.84 | 0.48 |
| *Mamba-based diffusion models.* | | | | | | | |
| DiM-H [79] | Conv + SSM | 708 | 7.39 | 3.78 | - | - | - |
| DiS-H/2 [25] | Conv + SSM | - | 8.59 | 2.88 | 272.33 | 0.84 | 0.56 |
| DiffuSSM-XL [91] | Attn + SSM | 1066.2 | - | 3.41 | 255.06 | 0.85 | 0.49 |
| *Baselines and Ours (w/ the same hyperparameters).* | | | | | | | |
| DiT-XL/2 (1.3M) [62] | Attn | 524.70 | 18.58 | 13.78 | - | - | - |
| **DiCo-XL** (1.3M) | Conv | 349.78 | 57.45 | 8.10 | 132.85 | 0.78 | 0.62 |
| DiT-XL/2 (3M) | Attn | 524.70 | 18.58 | 12.03 | 105.25 | 0.75 | 0.64 |
| **DiCo-XL** (3M) | Conv | 349.78 | 57.45 | 7.48 | 146.35 | 0.78 | 0.63 |
| DiT-XL/2 (w/ CFG) | Attn | 524.70 | 18.58 | 3.04 | 240.82 | 0.84 | 0.54 |
| **DiCo-XL** (w/ CFG) | Conv | 349.78 | 57.45 | **2.53** | **275.74** | 0.83 | **0.56** |

We analyze both the overall architecture and the contributions of individual components within DiCo to better understand their impact on model performance.

**Architecture Ablation.** We evaluate the performance of DiCo under various architectural designs and conduct a fair comparison with DiT. As shown in Table 4, DiCo consistently outperforms DiT across all structures while also delivering significant efficiency gains. These results highlight the potential of DiCo as a strong and efficient alternative to DiT.

**Component-wise Ablation.** We conduct a component-wise analysis of DiCo, examining the effects of the activation function, convolutional kernel size, compact channel attention (CCA), and the conv module (CM). The overall ablation results are summarized in Table 5. Increasing the convolutional kernel size leads to further perfor-

Table 3: **Comparison for text-to-image generation** on MS-COCO. We follow the setup in [96].

| Model | Type | FID |
|---|---|---|
| AttnGAN [90] | GAN | 35.49 |
| DM-GAN [101] | GAN | 32.64 |
| VQ-Diffusion [27] | Discrete Diffusion | 19.75 |
| DF-GAN [78] | GAN | 19.32 |
| XMC-GAN [98] | GAN | 9.33 |
| Frido [24] | Diffusion | 8.97 |
| LAFITE [99] | GAN | 8.12 |
| U-Net [6] | Diffusion | 7.32 |
| U-ViT-S/2 [6] | Diffusion | 5.95 |
| U-ViT-S/2 (Deep) [6] | Diffusion | 5.48 |
| MMDiT [96] | Diffusion | 5.30 |
| MMDiT+REPA [96] | Diffusion | 4.14 |
| **DiCo-CrossAttn** | Diffusion | 4.87 |
| **DiCo-DWC** | Diffusion | 4.93 |

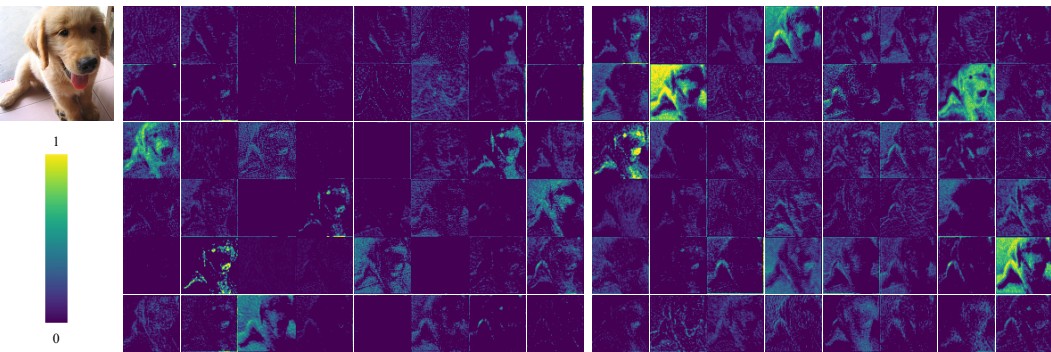

Figure 7: **CCA effectively reduces channel redundancy and enhances feature diversity.** *Left:* Features from the first stage of DiCo without CCA. *Right:* Features from the first stage of DiCo.

Table 4: **Ablation study on architectural design.** We compare different architectural variants, including isotropic, isotropic with skip connections, and U-shape. For all these architectural designs, our DiCo consistently outperforms DiT.

| Model | Skip | Hierarchy | #Params | Gflops | Throughput | FID↓ | IS↑ | Pre.↑ | Rec.↑ |
|---|---|---|---|---|---|---|---|---|---|
| DiT (*iso.*) [62] | ✗ | ✗ | 32.9M | 6.06 | 1086.81 | 67.16 | 20.41 | 0.37 | **0.57** |
| **DiCo (*iso.*)** | ✗ | ✗ | 33.7M | 5.67 | **1901.39** | **60.58** | **25.44** | **0.44** | 0.55 |
| DiT (*iso.&skip*) | ✓ | ✗ | 34.3M | 6.44 | 1037.55 | 62.63 | 22.08 | 0.39 | **0.56** |
| **DiCo (*iso.&skip*)** | ✓ | ✗ | 35.1M | 6.04 | **1807.22** | **56.95** | **26.84** | **0.46** | **0.56** |
| DiT (*U-shape*) | ✓ | ✓ | 33.0M | 4.23 | 1140.93 | 54.00 | 28.52 | 0.42 | **0.59** |
| **DiCo (*U-shape*)** | ✓ | ✓ | 33.1M | 4.25 | **1695.73** | **49.97** | **31.38** | **0.48** | 0.58 |

Table 5: **Ablation study on the components of DiCo.** ‡ indicates that the model depth and width are adjusted for a fair comparison. We analyze the effects of the activation function, DWC kernel size, compact channel attention (CCA), and the conv module (CM). For the CM, we compare it against several advanced efficient attention mechanisms to validate its effectiveness and efficiency.

| Model | #Params | Gflops | Throughput | FID↓ | IS↑ | Pre.↑ | Rec.↑ |
|---|---|---|---|---|---|---|---|
| **DiCo-S** | 33.1M | 4.25 | **1695.73** | 49.97 | 31.38 | 0.48 | 0.58 |
| GELU→ReLU | 33.1M | 4.25 | 1695.68 | 51.26 | 30.23 | 0.47 | 0.57 |
| 3×3→5×5 DWC | 33.2M | 4.29 | 1628.59 | 48.03 | 32.51 | 0.49 | 0.58 |
| 3×3→7×7 DWC | 33.4M | 4.34 | 1469.45 | **47.49** | **32.93** | **0.49** | **0.59** |
| **Compact Channel Attn (CCA)** | 33.1M | 4.25 | 1695.73 | **49.97** | **31.38** | **0.48** | 0.58 |
| w/o CCA ‡ | 33.0M | 4.24 | **1731.13** | 54.78 | 28.40 | 0.48 | 0.57 |
| CCA→SE module ‡ [40] | 32.9M | 4.25 | 1657.10 | 50.89 | 30.49 | 0.48 | 0.57 |
| CCA→Channel Self-Attn ‡ [97] | 33.2M | 4.26 | 1569.51 | 50.24 | 30.85 | 0.45 | **0.59** |
| **Conv Module (CM)** | 33.1M | 4.25 | **1695.73** | **49.97** | **31.38** | **0.48** | 0.58 |
| CM→Self-Attn ‡ [83] | 33.0M | 4.23 | 1140.93 | 54.00 | 28.52 | 0.42 | 0.59 |
| CM→Window Attn ‡ [56] | 33.0M | 4.34 | 1165.22 | 53.23 | 28.33 | 0.43 | 0.59 |
| CM→Focused Linear Attn ‡ [29] | 32.9M | 4.33 | 971.49 | 50.60 | 30.85 | 0.46 | **0.60** |
| CM→Agent Attn ‡ [65] | 33.3M | 4.24 | 1160.89 | 52.07 | 28.82 | 0.43 | **0.60** |

mance gains but at the expense of reduced efficiency, highlighting a trade-off between performance and computational cost.

The introduction of CCA results in a 4.81-point improvement in FID. As illustrated in Fig. 7, CCA significantly enhances feature diversity, demonstrating its effectiveness in improving the model's representational capacity. We also compare CCA with SE module [40] and Channel-wise Self-Attention [97]; despite its simplicity, CCA achieves superior performance and higher efficiency.

For the Conv Module, we benchmark it against several advanced efficient attention mechanisms (Window Attention [56], Focused Linear Attention [29], Agent Attention [65]). The results show that our CM offers both better performance and computational efficiency.

## 5 Conclusion

We propose a new backbone for diffusion models, Diffusion ConvNet (DiCo), as a compelling alternative to the Diffusion Transformer (DiT). DiCo replaces self-attention with a combination of $1 \times 1$ pointwise convolutions and $3 \times 3$ depthwise convolutions, and incorporates a compact channel attention mechanism to reduce channel redundancy and enhance feature diversity. As a fully convolutional network, DiCo surpasses recent diffusion models on the ImageNet 256×256 and 512×512 benchmarks, while achieving significant efficiency gains. Furthermore, the purely convolutional DiCo demonstrates strong potential in text-to-image generation. We look forward to further scaling up DiCo and extending it to broader generative tasks.

# Acknowledgements

This research is partially funded by National Natural Science Foundation of China (Grant No. 62576342), Beijing Natural Science Foundation (4252054), Youth Innovation Promotion Association CAS(Grant No.2022132), Beijing Nova Program(20230484276).

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

# Appendix

We provide the following supplementary information in the Appendix:

- Sec. A. detailed configurations of DiCo model variants.
- Sec. B. additional implementation details of DiCo.
- Sec. C. scalability analysis of DiCo models.
- Sec. D. additional comparison results with generative model family.
- Sec. E. additional samples generated by DiCo-XL models.
- Sec. F. discussion of limitations of this work.
- Sec. G. discussion of broader impacts of this work.

## A   DiCo Model Variants

We introduce several variants of DiCo, each scaled to different model sizes. Specifically, we present five variants: Small (33M parameters), Base (130M), Large (460M), XLarge (700M), and Huge (1B). These variants are created by adjusting the hidden size $D$, the number of layers ($L_1, L_2, L_3, L_4, L_5$), and the FFN ratio. They span a wide range of parameter counts and FLOPs, from 33M to 1B parameters and from 4.25 Gflops to 194.15 Gflops, offering a comprehensive foundation for evaluating scalability and efficiency. Notably, when compared to their corresponding DiT counterparts, our DiCo models require only 70.1% to 74.6% of the FLOPs, demonstrating the computational efficiency of our design. The detailed configurations for these variants are provided in Table 6.

Table 6: **Architecture variants of DiCo.** Gflops are measured with an input size of $32 \times 32 \times 4$. Compared to DiT, our DiCo models are more computationally efficient.

| Model | #Params (M) | Gflops | $\frac{\text{Gflops}_{\text{DiCo}}}{\text{Gflops}_{\text{DiT}}}$ | Hidden Size $D$ | #Layers | FFN Ratio |
|---|---|---|---|---|---|---|
| DiCo-S | 33.1 | 4.25 | 70.1% | 128 | [5, 4, 4, 4, 4] | 2 |
| DiCo-B | 130.0 | 16.88 | 73.3% | 256 | [5, 4, 4, 4, 4] | 2 |
| DiCo-L | 463.9 | 60.24 | 74.6% | 352 | [9, 8, 9, 8, 9] | 2 |
| DiCo-XL | 701.2 | 87.30 | 73.6% | 416 | [9, 9, 10, 9, 9] | 2 |
| DiCo-H | 1037.4 | 194.15 | - | 416 | [14, 12, 10, 12, 14] | 4 |

Table 7: **Implementation details of DiCo variants.** For DiCo-S/B/L/XL, we follow the same experimental settings as DiT [6]. For the largest variant DiCo-H, we adopt the training hyperparameters from U-ViT [6], increasing the batch size and learning rate to accelerate training.

| Model | DiCo-S | DiCo-B | DiCo-L | DiCo-XL | DiCo-XL | DiCo-H |
|---|---|---|---|---|---|---|
| Resolution | $256 \times 256$ | $256 \times 256$ | $256 \times 256$ | $256 \times 256$ | $512 \times 512$ | $256 \times 256$ |
| Optimizer | AdamW | AdamW | AdamW | AdamW | AdamW | AdamW |
| Betas | $(0.9, 0.999)$ | $(0.9, 0.999)$ | $(0.9, 0.999)$ | $(0.9, 0.999)$ | $(0.9, 0.999)$ | $(0.99, 0.99)$ |
| Weight decay | 0 | 0 | 0 | 0 | 0 | 0 |
| Peak learning rate | $1 \times 10^{-4}$ | $1 \times 10^{-4}$ | $1 \times 10^{-4}$ | $1 \times 10^{-4}$ | $1 \times 10^{-4}$ | $2 \times 10^{-4}$ |
| Learning rate schedule | constant | constant | constant | constant | constant | constant |
| Warmup steps | 0 | 0 | 0 | 0 | 0 | 5K |
| Global batch size | 256 | 256 | 256 | 256 | 256 | 1024 |
| Numerical precision | fp32 | fp32 | fp32 | fp32 | fp32 | fp32 |
| Training steps | 400K | 400K | 400K | 3750K | 3000K | 1000K |
| Computational resources | 8 A100 | 8 A100 | 16 A100 | 32 A100 | 64 A100 | 64 A100 |
| Training Time | 11 hours | 16 hours | 29 hours | 266 hours | 256 hours | 113 hours |
| Data Augmentation | random flip | random flip | random flip | random flip | random flip | random flip |
| VAE | sd-ft-ema | sd-ft-ema | sd-ft-ema | sd-ft-ema | sd-ft-ema | sd-ft-ema |
| Sampler | iDDPM | iDDPM | iDDPM | iDDPM | iDDPM | iDDPM |
| Sampling steps | 250 | 250 | 250 | 250 | 250 | 250 |

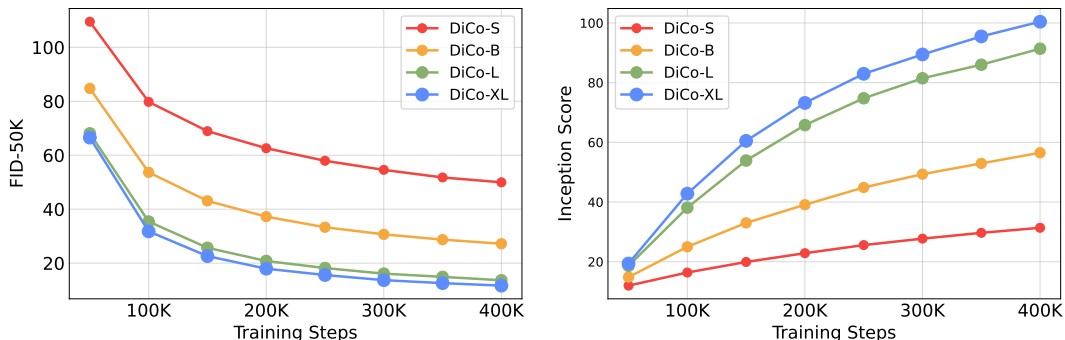

Figure 8: **Scaling the DiCo models consistently improves performance throughout training.** We report FID-50K and Inception Score across training iterations for four DiCo variants.

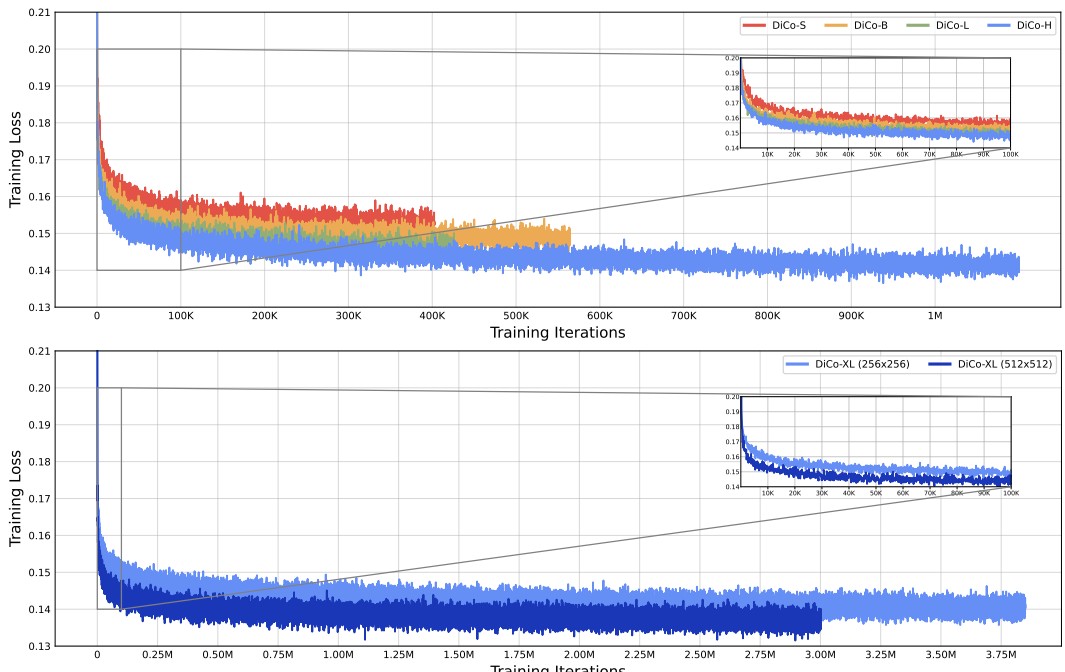

Figure 9: **Training loss curves for DiCo models.** We also highlight early training dynamics during the first 100K iterations. Larger DiCo variants exhibit lower training losses, reflecting improved optimization with scale.

## B   Additional Implementation Details

For DiCo-S/B/L/XL models, we adopt the same experimental settings as those used for DiT. For DiCo-H, the largest variant, we use the hyperparameters from U-ViT, increasing the batch size and learning rate to expedite training. All experiments are conducted on NVIDIA A100 (80G) GPUs. During inference, all models use the iDDPM [60] sampler with 250 sampling steps. The whole implementation details are summarized in Table 7.

## C   Scalability Analysis

**Impact of scaling on metrics.** Table 8 and Fig. 8 illustrate the impact of DiCo model scaling across various evaluation metrics. Our results show that scaling DiCo consistently enhances performance across all metrics throughout training, highlighting its potential as a strong candidate for a large-scale foundational diffusion model.

Table 8: **Performance of DiCo models without CFG at different training steps on ImageNet 256×256.** Scaling the ConvNet backbone consistently leads to improved generative performance.

| Model | Gflops | Training Steps | FID ↓ | sFID ↓ | IS ↑ | Precision ↑ | Recall ↑ |
|---|---|---|---|---|---|---|---|
| DiCo-S | 4.25 | 50K | 109.50 | 18.28 | 12.00 | 0.283 | 0.322 |
| | | 100K | 79.79 | 15.60 | 16.39 | 0.382 | 0.440 |
| | | 150K | 68.93 | 13.98 | 19.93 | 0.411 | 0.498 |
| | | 200K | 62.61 | 13.18 | 22.87 | 0.433 | 0.531 |
| | | 250K | 57.95 | 12.47 | 25.58 | 0.450 | 0.546 |
| | | 300K | 54.58 | 11.99 | 27.74 | 0.467 | 0.565 |
| | | 350K | 51.77 | 11.69 | 29.68 | 0.479 | 0.576 |
| | | 400K | **49.97** | **11.41** | **31.38** | **0.481** | **0.582** |
| DiCo-B | 16.88 | 50K | 84.78 | 14.24 | 14.85 | 0.392 | 0.423 |
| | | 100K | 53.66 | 9.13 | 25.01 | 0.494 | 0.544 |
| | | 150K | 43.09 | 8.21 | 33.02 | 0.536 | 0.571 |
| | | 200K | 37.24 | 8.03 | 39.10 | 0.561 | 0.596 |
| | | 250K | 33.31 | 7.72 | 44.89 | 0.577 | 0.598 |
| | | 300K | 30.68 | 7.59 | 49.34 | 0.584 | 0.597 |
| | | 350K | 28.72 | 7.55 | 52.95 | 0.594 | 0.599 |
| | | 400K | **27.20** | **7.43** | **56.52** | **0.603** | **0.617** |
| DiCo-L | 60.24 | 50K | 68.11 | 11.50 | 18.62 | 0.480 | 0.465 |
| | | 100K | 35.42 | 6.52 | 38.12 | 0.607 | 0.561 |
| | | 150K | 25.69 | 6.04 | 53.91 | 0.643 | 0.579 |
| | | 200K | 20.81 | 5.80 | 65.78 | 0.665 | 0.589 |
| | | 250K | 18.19 | 5.70 | 74.77 | 0.676 | 0.588 |
| | | 300K | 16.11 | 5.60 | 81.47 | 0.685 | 0.594 |
| | | 350K | 14.92 | 5.58 | 86.01 | 0.689 | 0.602 |
| | | 400K | **13.66** | **5.50** | **91.37** | **0.694** | **0.604** |
| DiCo-XL | 87.30 | 50K | 66.53 | 11.35 | 19.41 | 0.501 | 0.472 |
| | | 100K | 31.78 | 6.39 | 42.81 | 0.637 | 0.563 |
| | | 150K | 22.61 | 5.94 | 60.49 | 0.672 | 0.572 |
| | | 200K | 17.95 | 5.67 | 73.21 | 0.693 | 0.582 |
| | | 250K | 15.60 | 5.58 | 82.96 | 0.697 | 0.591 |
| | | 300K | 13.70 | 5.52 | 89.44 | 0.707 | 0.599 |
| | | 350K | 12.59 | 5.48 | 95.49 | 0.710 | 0.605 |
| | | 400K | **11.67** | **5.38** | **100.42** | **0.711** | **0.606** |

**Impact of scaling on training loss.** We further analyze the effect of model scale on training loss. As shown in Fig. 9, larger DiCo models consistently achieve lower training losses, indicating more effective optimization with increased scale.

# D  Additional Comparison Results

**ImageNet 256×256.** Table 9 presents a comparison across generative model family on ImageNet 256×256. Among diffusion models, our DiCo-XL achieves a strong FID of 2.05 with only 87.3 Gflops. Our largest variant, DiCo-H, attains an FID of 1.90. When compared to other generative model types, DiCo also demonstrates competitive performance. Notably, DiCo-H, with just 1B parameters, outperforms VAR-d30—which has 2B parameters—in terms of FID.

**ImageNet 512×512.** Table 10 presents the results on ImageNet 512×512. Among diffusion models, our DiCo-XL achieves an FID of 2.53 with only 349.8 GFLOPs. Compared to other generative models, DiCo continues to demonstrate strong performance. Specifically, DiCo-XL, with only 701M parameters, outperforms VAR-d36-s, which has 2.3B parameters, achieving superior FID performance with significantly fewer parameters.

Table 9: **Comparison with generative model family on ImageNet 256×256.** We report the performance of state-of-the-art generative models across different paradigms, including GAN-based, masked prediction (Mask.)-based, autoregressive (AR), visual-autoregressive (VAR), and diffusion (Diff.)-based models.

| Type | Model | #Params | Gflops | FID ↓ | IS ↑ | Precision ↑ | Recall ↑ |
|------|-------|---------|--------|-------|------|-------------|----------|
| GAN | BigGAN-deep [8] | 160M | - | 6.95 | 171.4 | 0.87 | 0.28 |
| GAN | GigaGAN [43] | 569M | - | 3.45 | 225.5 | 0.84 | 0.61 |
| GAN | StyleGAN-XL [72] | 166M | - | 2.30 | 265.1 | 0.78 | 0.53 |
| Mask. | MaskGIT [11] | 227M | - | 6.18 | 182.1 | 0.80 | 0.51 |
| Mask. | RCG [51] | 502M | - | 3.49 | 215.5 | - | - |
| Mask. | TiTok-S-128 [95] | 287M | - | 1.97 | 281.8 | - | - |
| AR | VQ-GAN-re [21] | 1.4B | - | 5.20 | 280.3 | - | - |
| AR | ViTVQ-re [94] | 1.7B | - | 3.04 | 227.4 | - | - |
| AR | RQTran.-re [50] | 3.8B | - | 3.80 | 323.7 | - | - |
| AR | LLamaGen-3B [77] | 3.1B | - | 2.18 | 263.3 | 0.81 | 0.58 |
| AR | MAR-H [52] | 943M | - | 1.55 | 303.7 | 0.81 | 0.62 |
| AR | PAR-3B [86] | 3.1B | - | 2.29 | 255.5 | 0.82 | 0.58 |
| AR | RandAR-XXL [61] | 1.4B | - | 2.15 | 322.0 | 0.79 | 0.62 |
| VAR | VAR-d16 [80] | 310M | - | 3.30 | 274.4 | 0.84 | 0.51 |
| VAR | VAR-d20 | 600M | - | 2.57 | 302.6 | 0.83 | 0.56 |
| VAR | VAR-d24 | 1.0B | - | 2.09 | 312.9 | 0.82 | 0.59 |
| VAR | VAR-d30 | 2.0B | - | 1.92 | 323.1 | 0.82 | 0.59 |
| Diff. | ADM-U [14] | 608M | 742 | 3.94 | 215.8 | 0.83 | 0.53 |
| Diff. | U-ViT-L/2 [6] | 287M | 77 | 3.40 | 219.9 | 0.83 | 0.52 |
| Diff. | U-ViT-H/2 [6] | 501M | 133.3 | 2.29 | 263.9 | 0.82 | 0.57 |
| Diff. | Simple Diffusion [37] | 2.0B | - | 2.77 | 211.8 | - | - |
| Diff. | VDM++ [47] | 2.0B | - | 2.12 | 267.7 | - | - |
| Diff. | DiT-XL/2 [62] | 675M | 118.7 | 2.27 | 278.2 | 0.83 | 0.57 |
| Diff. | SiT-XL [58] | 675M | 118.7 | 2.06 | 277.5 | 0.83 | 0.59 |
| Diff. | DiM-H [79] | 860M | 210 | 2.21 | - | - | - |
| Diff. | DiS-H/2 [25] | 901M | - | 2.10 | 271.3 | 0.82 | 0.58 |
| Diff. | DiffuSSM-XL [91] | 673M | 280.3 | 2.28 | 259.1 | 0.86 | 0.56 |
| Diff. | DiMSUM-L/2 [63] | 460M | 84.49 | 2.11 | - | - | 0.59 |
| Diff. | DiG-XL/2 [100] | 676M | 89.40 | 2.07 | 279.0 | 0.82 | 0.60 |
| Diff. | DiC-H [81] | 1.0B | 204.4 | 2.25 | - | - | - |
| Diff. | **DiCo-XL** | 701M | 87.30 | 2.05 | 282.2 | 0.83 | 0.59 |
| Diff. | **DiCo-H** | 1.0B | 194.15 | 1.90 | 284.3 | 0.83 | 0.61 |

**ImageNet 1024×1024.** We conduct experiments on ImageNet 1024×1024. Since the original DiT paper does not report results at this resolution, we train both DiT and DiCo from scratch under the same settings for a fair comparison. We use small-scale models and train them for 400K iterations. Table 11 clearly shows that DiCo scales much more efficiently to high resolutions—achieving a 5× speedup with better generation quality.

**Comparison with FlowDCN.** Deformable convolution is a strong and widely used convolutional variant, and FlowDCN [85] represents an important baseline in this space. To fairly compare with FlowDCN, we retain its original isotropic architecture and training setup, and replace its MultiScale DCN with our proposed Conv Module. Table 12 shows that our Conv Module achieves better generative performance and 2.1× higher throughput compared to MultiScale DCN, despite being structurally simpler and easier to implement.

**Comparison with Modern DiTs.** Given the rapid advancements in modern DiTs, it is important to compare our method with recent DiTs such as LightningDiT [93]. We find that after aligning with their experimental settings, our method achieves competitive performance along with a clear efficiency advantage. Table 13 shows that our model achieves an FID score of 1.33 and offers a 2.8×

Table 10: **Comparison with generative model family on ImageNet 512×512.** We report the performance of state-of-the-art generative models across different paradigms, including GAN-based, masked prediction (Mask.)-based, autoregressive (AR), visual-autoregressive (VAR), and diffusion (Diff.)-based models.

| Type | Model | #Params | Gflops | FID ↓ | IS ↑ | Precision ↑ | Recall ↑ |
|------|-------|---------|--------|-------|------|-------------|----------|
| GAN | BigGAN-deep [8] | 160M | - | 8.43 | 177.9 | 0.88 | 0.29 |
| GAN | StyleGAN-XL [72] | 166M | - | 2.41 | 267.8 | 0.77 | 0.52 |
| Mask. | MaskGIT [11] | 227M | - | 7.32 | 156.0 | 0.78 | 0.50 |
| AR | VQ-GAN [21] | 227M | - | 26.52 | 66.8 | 0.73 | 0.31 |
| VAR | VAR-d36-s [80] | 2.3B | - | 2.63 | 303.2 | - | - |
| Diff. | ADM-U [14] | 731M | 2813 | 3.85 | 221.7 | 0.84 | 0.53 |
| Diff. | U-ViT-L/4 [6] | 287M | 76.5 | 4.67 | 213.3 | 0.87 | 0.45 |
| Diff. | U-ViT-H/4 [6] | 501M | 133.3 | 4.05 | 263.8 | 0.84 | 0.48 |
| Diff. | Simple Diffusion [37] | 2.0B | - | 4.53 | 205.3 | - | - |
| Diff. | VDM++ [47] | 2.0B | - | 2.65 | 278.1 | - | - |
| Diff. | DiT-XL/2 [62] | 675M | 524.7 | 3.04 | 240.8 | 0.84 | 0.54 |
| Diff. | SiT-XL [58] | 675M | 524.7 | 2.62 | 252.2 | 0.84 | 0.57 |
| Diff. | DiM-H [79] | 860M | 708 | 3.78 | - | - | - |
| Diff. | DiS-H/2 [25] | 901M | - | 2.88 | 272.3 | 0.84 | 0.56 |
| Diff. | DiffuSSM-XL [91] | 673M | 1066.2 | 3.41 | 255.1 | 0.85 | 0.49 |
| Diff. | **DiCo-XL** | 701M | 349.8 | 2.53 | 275.7 | 0.83 | 0.56 |

Table 11: **Comparison with DiT on ImageNet 1024×1024.** The performance is reported at 400K iterations without CFG.

| Model | #Params | FLOPs | Throughput ↑ | FID ↓ | IS ↑ | Precision ↑ | Recall ↑ |
|-------|---------|-------|--------------|-------|------|-------------|----------|
| DiT-S/2 | 33M | 241.8G | 28 it/s | 113.06 | 12.60 | 0.25 | 0.37 |
| **DiCo-S** | 33M | **67.8G** | **143 it/s** | **102.60** | **16.45** | **0.33** | **0.38** |

throughput gain over modern DiTs. This demonstrates that our convolutional backbone can match or exceed the generative quality of modern DiTs while being significantly more efficient.

# E   Model Samples

We present samples generated by the DiCo-XL models at resolutions of 256×256 and 512×512. Fig. 10 through 29 display uncurated samples across various classifier-free guidance scales and input class labels. As illustrated, our DiCo models demonstrate the ability to generate high-quality, detail-rich images.

# F   Limitations

While this work demonstrates the strong performance and scalability of DiCo through extensive experiments, there are some limitations to note. Due to limited computational resources, our model is scaled to 1B parameters, while some advanced generative models have been scaled to even larger sizes. We aim to investigate the broader generative potential of DiCo in future research.

# G   Broader Impacts

Our work on the generative model DiCo contributes to advances in controllable image generation. This has potential positive applications in data augmentation, scientific visualization, and accessibility technologies. However, such models may also be misused to generate misleading or harmful content,

Table 12: **Comparison with FlowDCN on ImageNet 256×256.** The performance is reported at 400K iterations without CFG.

| Model | #Params | FLOPs | Throughput ↑ | FID ↓ | IS ↑ |
|---|---|---|---|---|---|
| SiT-S/2 [58] | 32.9M | 6.1G | 1234 it/s | 57.6 | 24.8 |
| FlowDCN-S/2 [85] | 30.3M | **4.4G** | 1194 it/s | 54.6 | 26.4 |
| **Ours-S/2** | 31.3M | 4.6G | **2489 it/s** | **52.1** | **29.2** |

Table 13: **Comparison with modern DiTs on ImageNet 256×256.** We follow the setup in [93].

| Model | Epochs | #Params | FLOPs | Throughput ↑ | FID ↓ | IS ↑ |
|---|---|---|---|---|---|---|
| REPA [96] | 800 | 675M | 118.7G | 77 it/s | 1.42 | 305.7 |
| LightningDiT [93] | 800 | 675M | 118.8G | 73 it/s | 1.35 | 295.3 |
| **Ours** | 800 | 679M | **118.3G** | **201 it/s** | **1.33** | **300.2** |

especially in contexts involving deepfakes or biased representations of specific classes. To mitigate these risks, we encourage responsible usage aligned with ethical AI guidelines and emphasize the importance of transparency when deploying generative models in real-world applications.

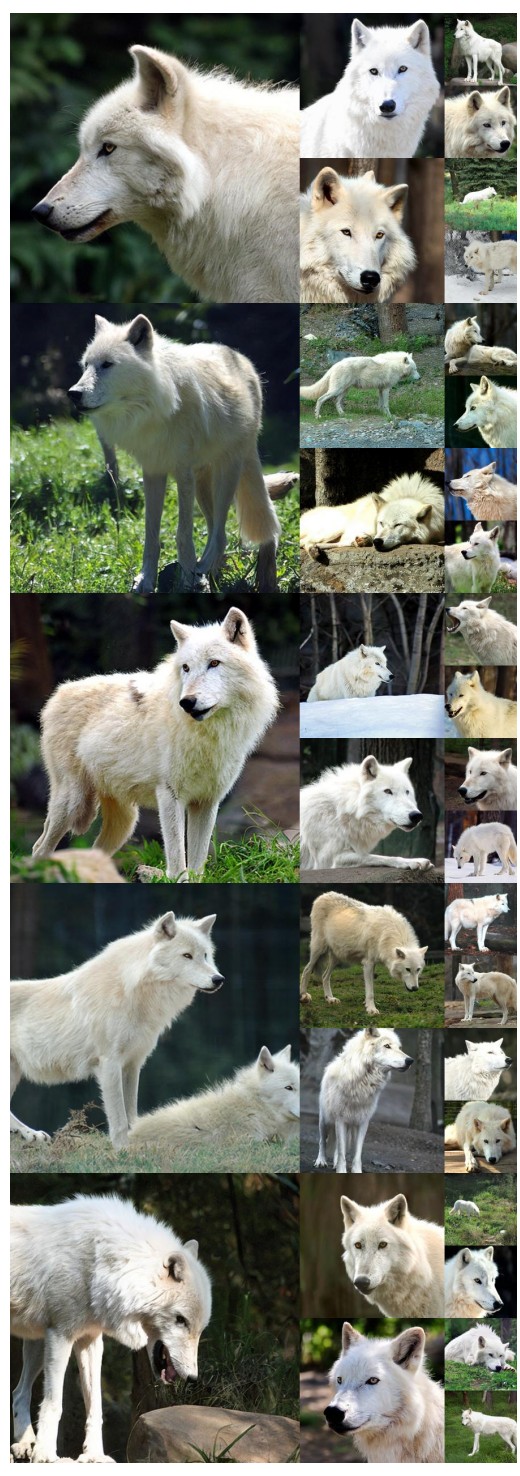

Figure 10: **Uncurated** $512 \times 512$ **DiCo-XL samples.**
Classifier-free guidance scale = 4.0
Class label = "arctic wolf" (270)

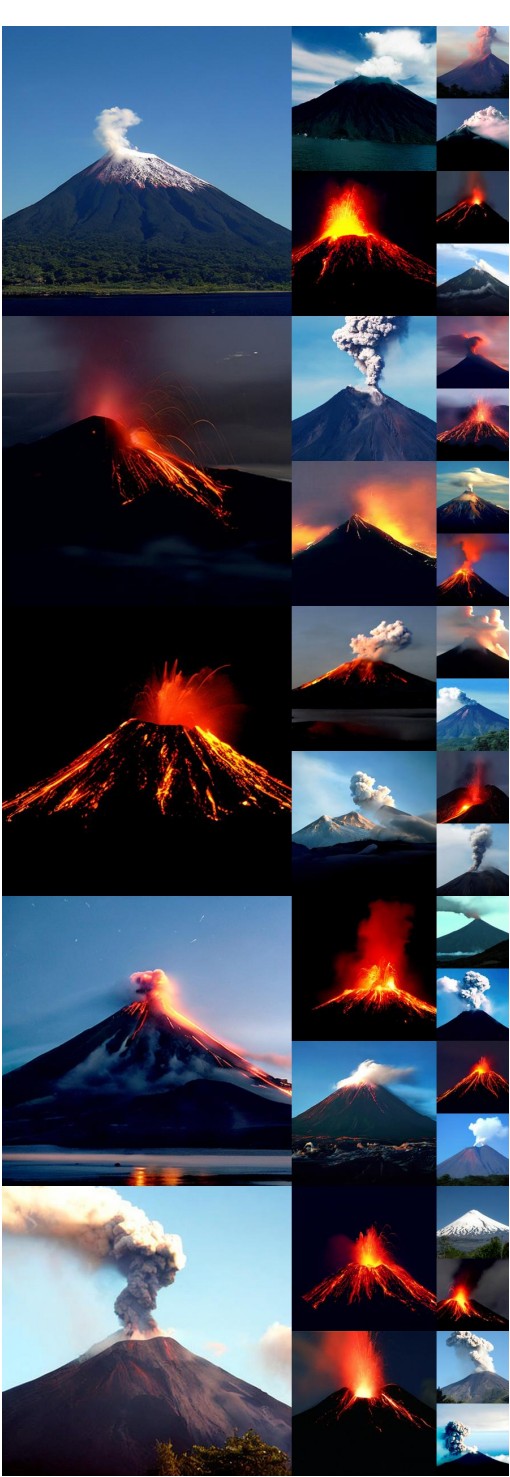

Figure 11: **Uncurated** $512 \times 512$ **DiCo-XL samples.**
Classifier-free guidance scale = 4.0
Class label = "volcano" (980)

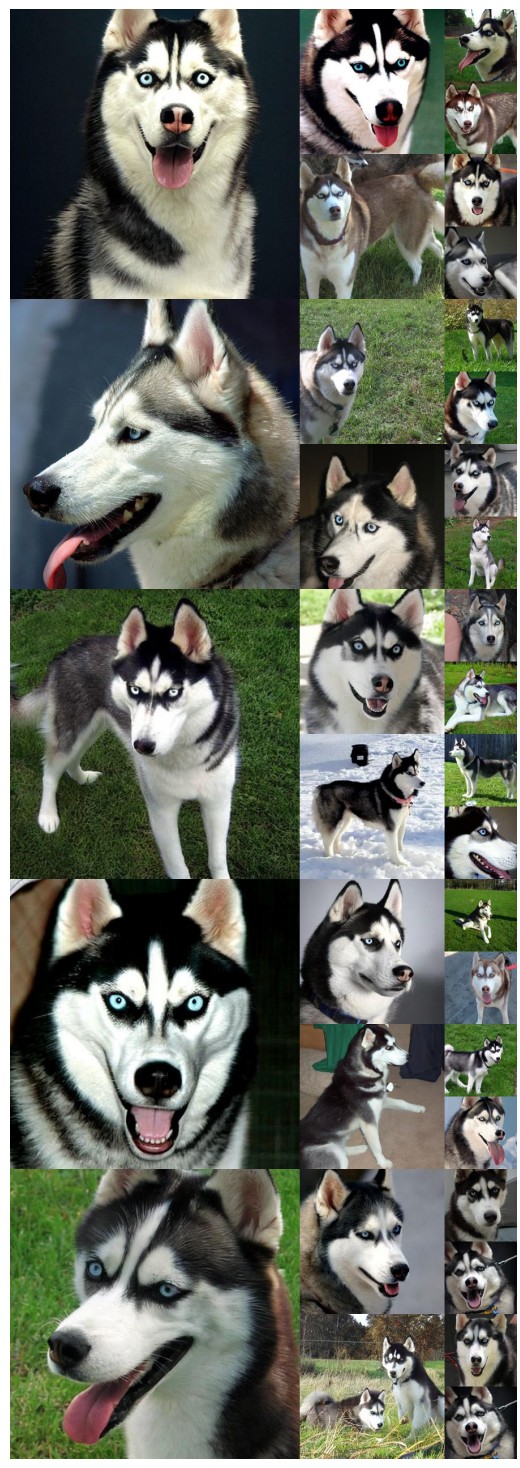

Figure 12: **Uncurated** $512\times512$ **DiCo-XL samples.**
Classifier-free guidance scale = 4.0
Class label = "husky" (250)

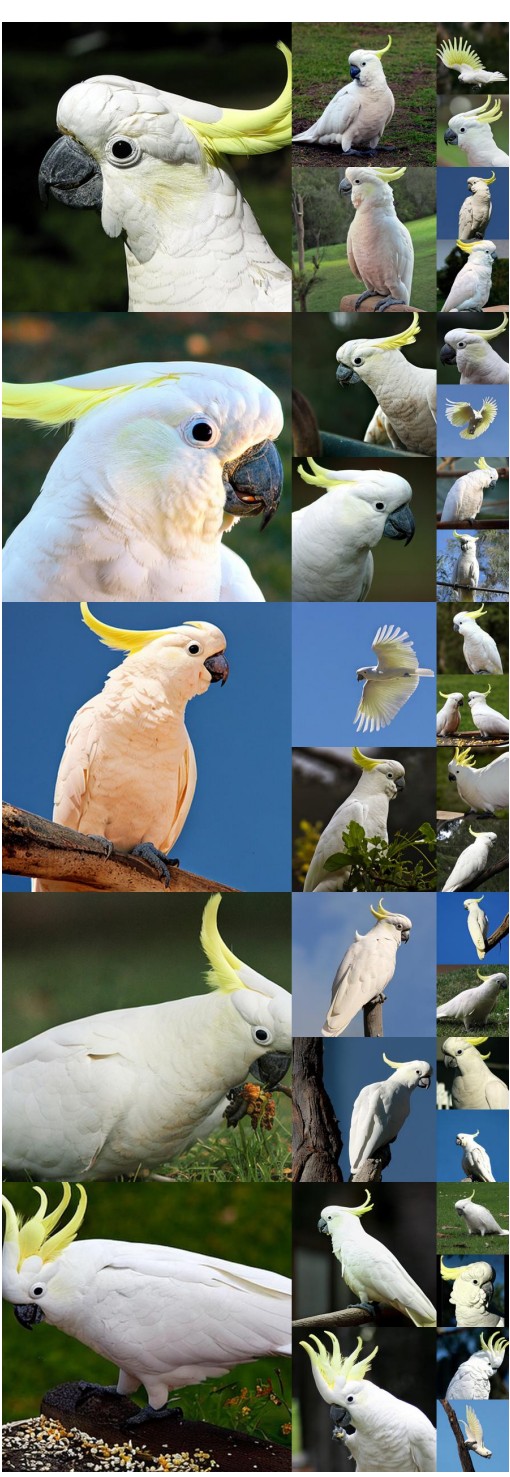

Figure 13: **Uncurated** $512\times512$ **DiCo-XL samples.**
Classifier-free guidance scale = 4.0
Class label = "ulphur-crested cockatoo" (89)

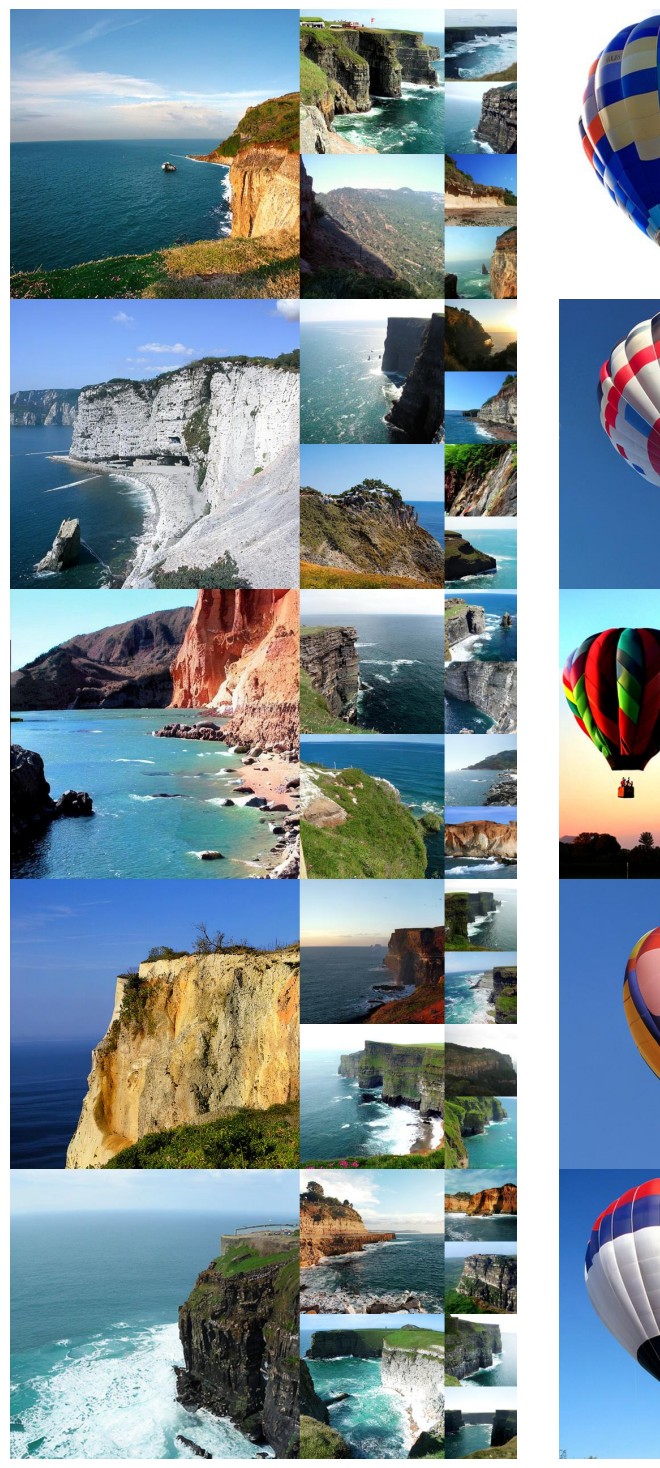

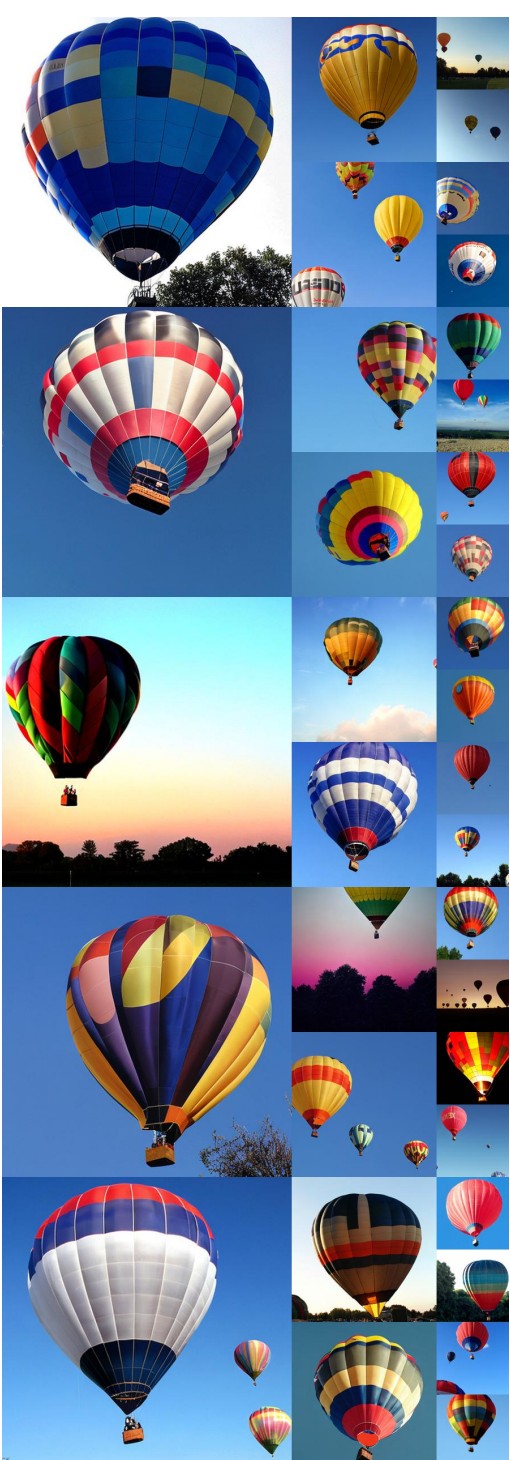

Figure 14: **Uncurated** $512 \times 512$ **DiCo-XL samples.**
Classifier-free guidance scale = 4.0
Class label = "cliff drop-off" (972)

Figure 15: **Uncurated** $512 \times 512$ **DiCo-XL samples.**
Classifier-free guidance scale = 4.0
Class label = "balloon" (417)

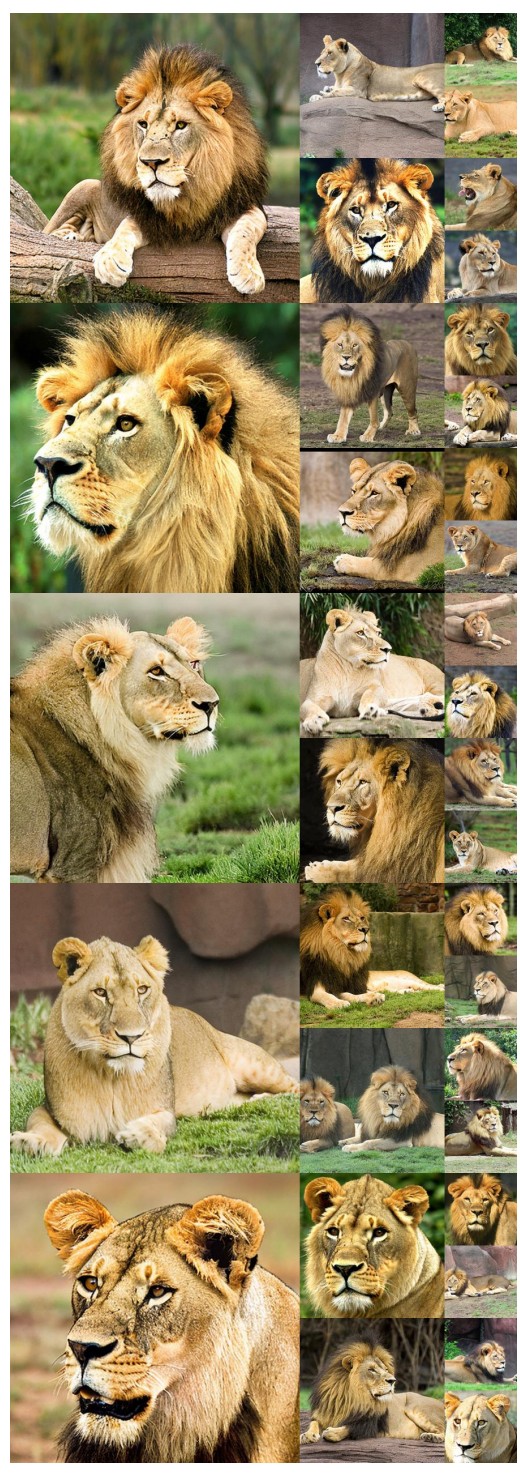

Figure 16: **Uncurated** $512 \times 512$ **DiCo-XL samples.**
Classifier-free guidance scale = 4.0
Class label = "lion" (291)

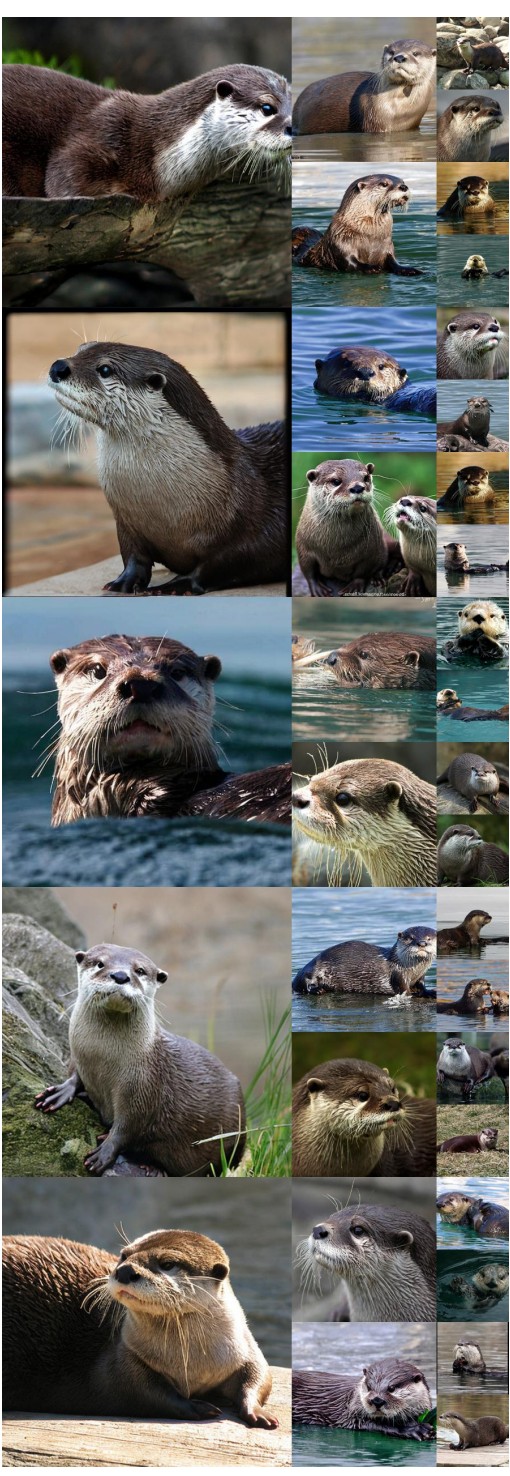

Figure 17: **Uncurated** $512 \times 512$ **DiCo-XL samples.**
Classifier-free guidance scale = 4.0
Class label = "otter" (360)

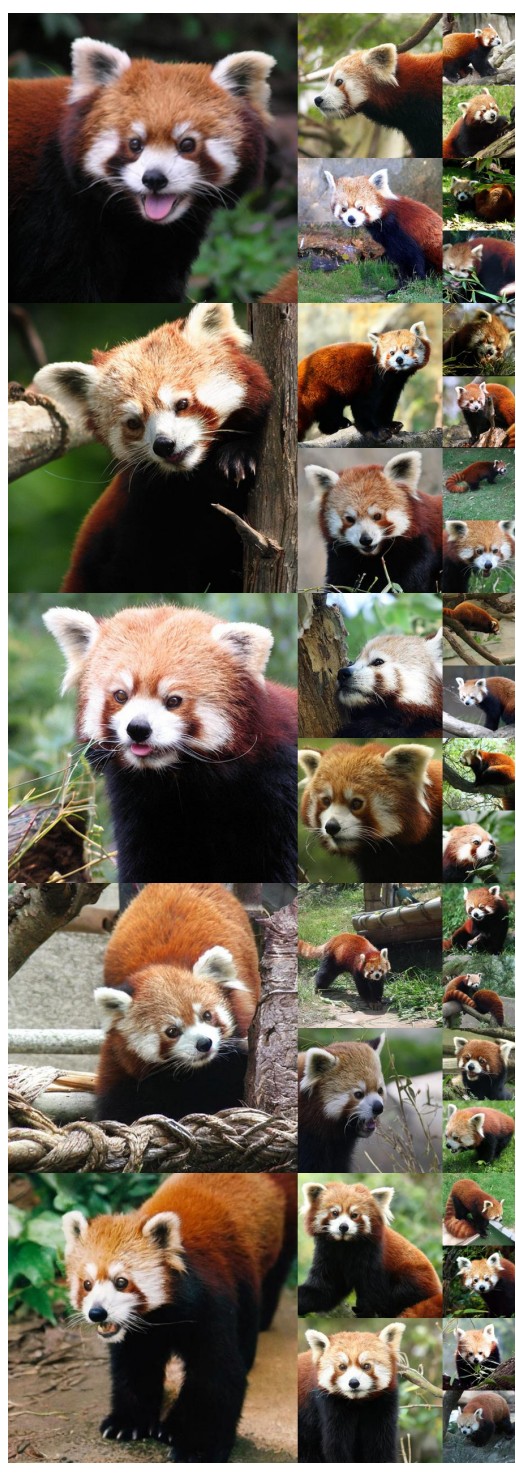

Figure 18: **Uncurated** $512 \times 512$ **DiCo-XL samples.**
Classifier-free guidance scale = 2.0
Class label = "red panda" (387)

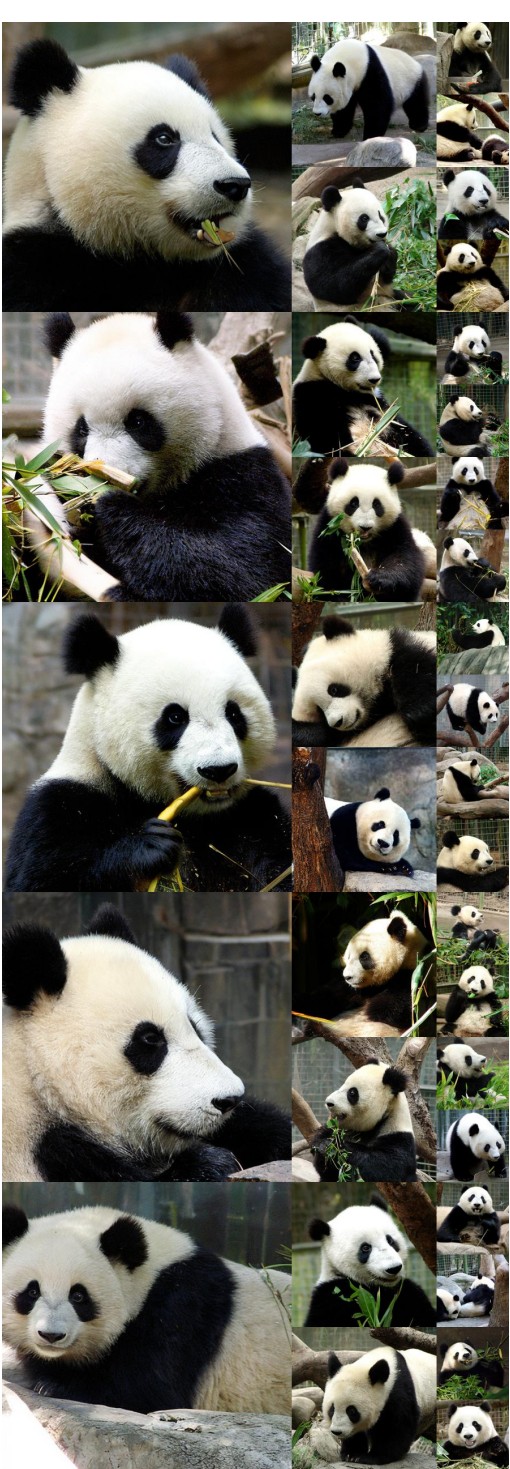

Figure 19: **Uncurated** $512 \times 512$ **DiCo-XL samples.**
Classifier-free guidance scale = 2.0
Class label = "panda" (388)

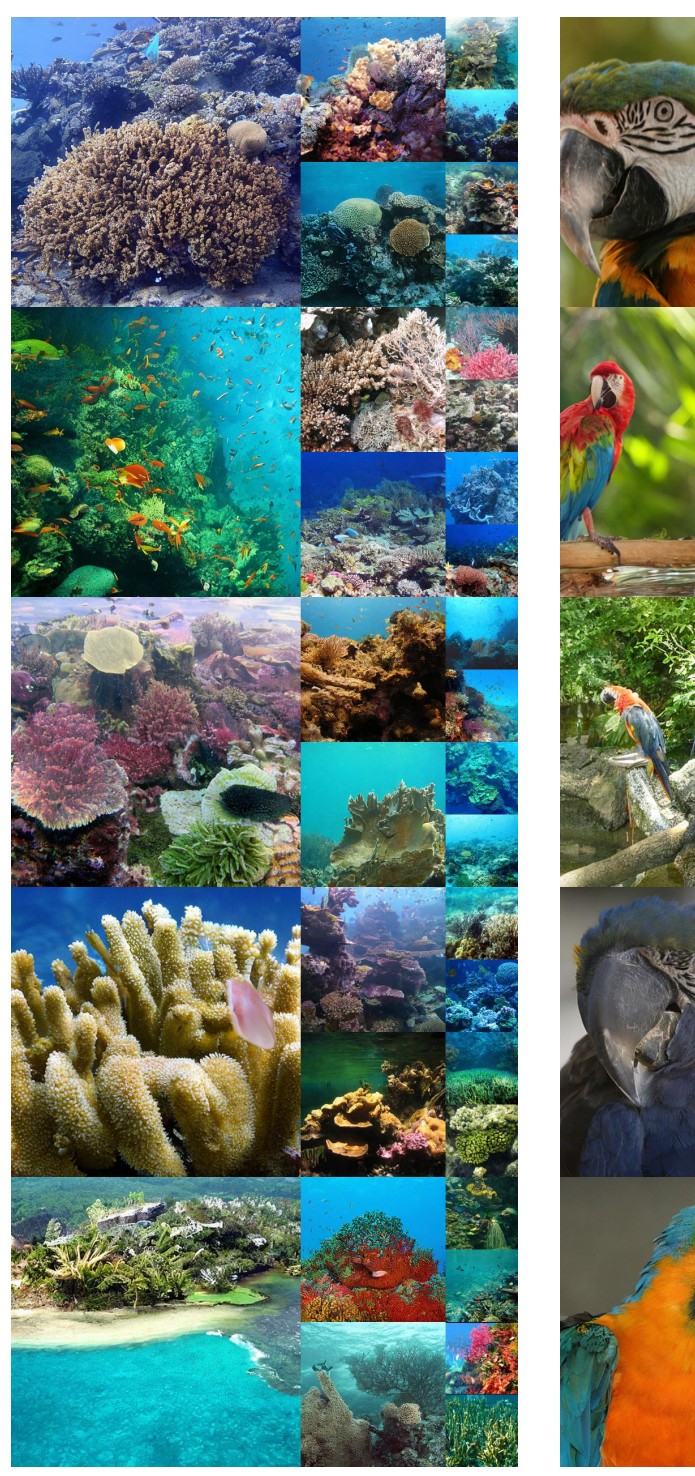

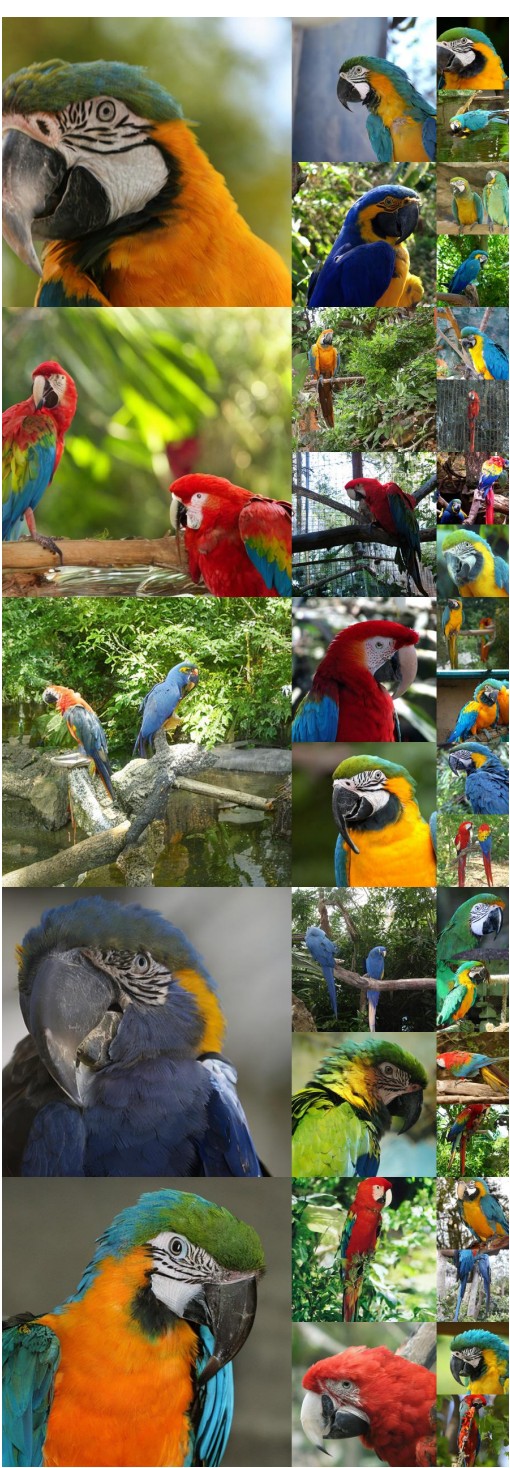

Figure 20: **Uncurated** $512 \times 512$ **DiCo-XL samples.**
Classifier-free guidance scale = 1.5
Class label = "coral reef" (973)

Figure 21: **Uncurated** $512 \times 512$ **DiCo-XL samples.**
Classifier-free guidance scale = 1.5
Class label = "macaw" (88)

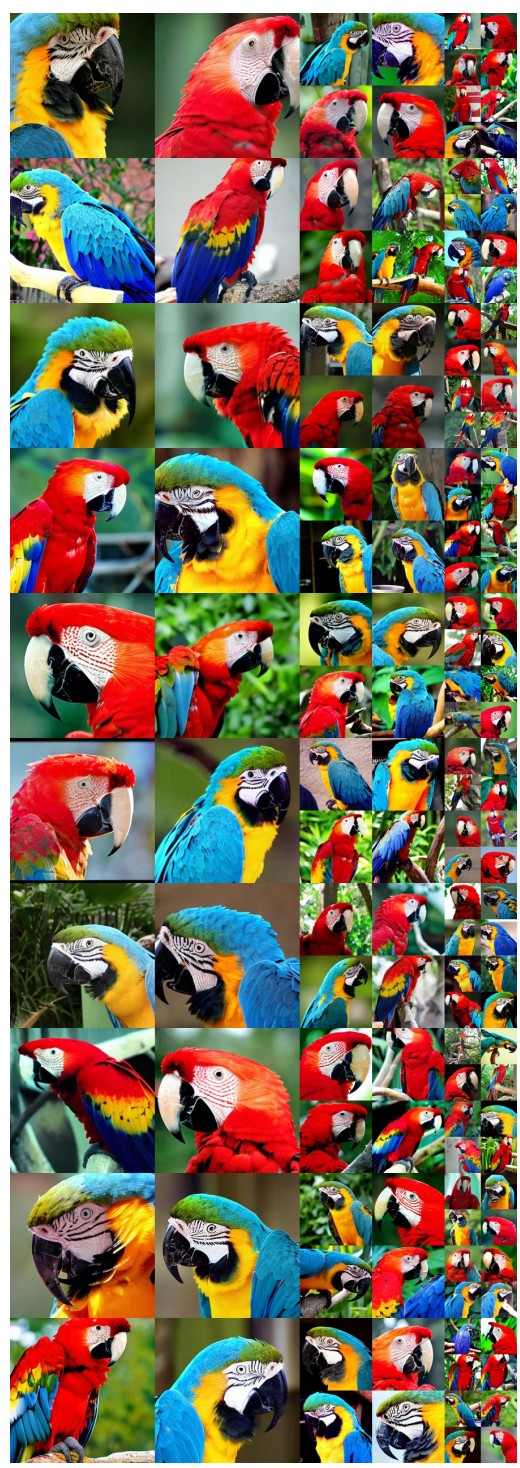

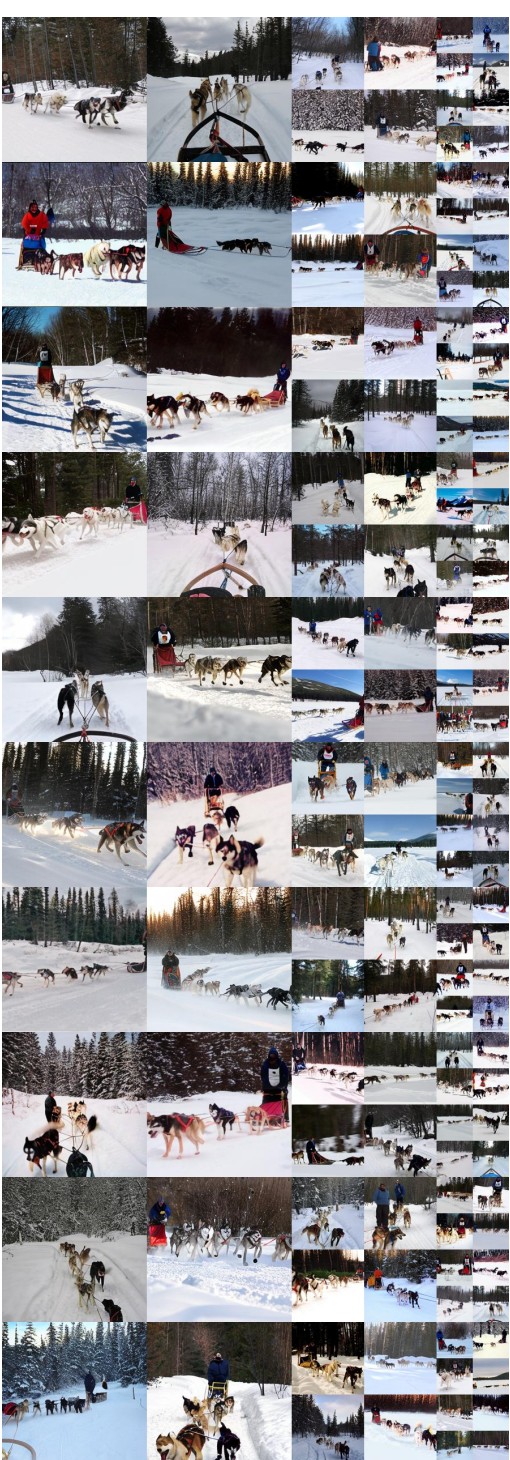

Figure 22: **Uncurated** $256 \times 256$ **DiCo-XL samples.**
Classifier-free guidance scale = 4.0
Class label = "macaw" (88)

Figure 23: **Uncurated** $256 \times 256$ **DiCo-XL samples.**
Classifier-free guidance scale = 4.0
Class label = "dog sled" (537)

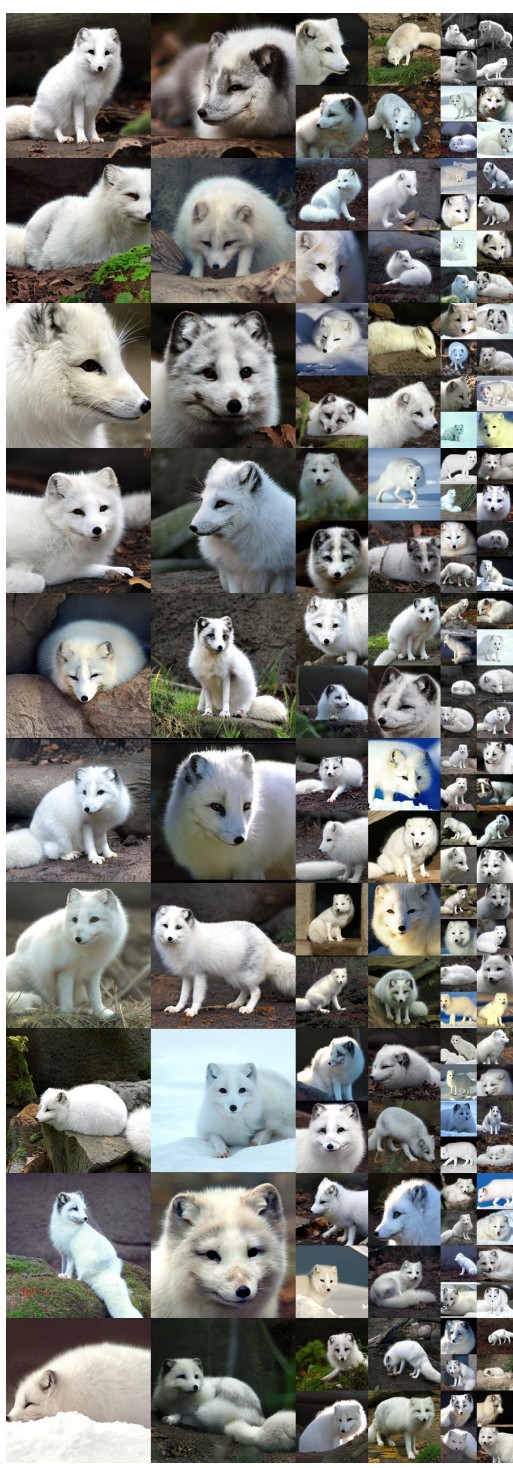

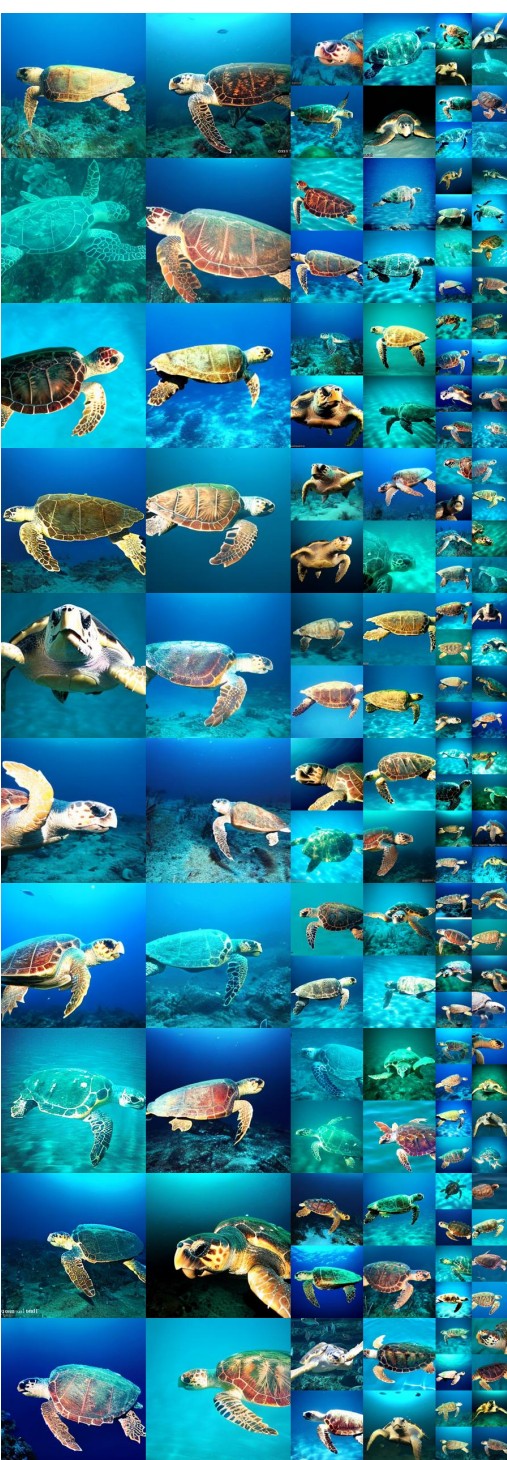

Figure 24: **Uncurated** $256 \times 256$ **DiCo-XL samples.**
Classifier-free guidance scale = 4.0
Class label = "arctic fox" (279)

Figure 25: **Uncurated** $256 \times 256$ **DiCo-XL samples.**
Classifier-free guidance scale = 4.0
Class label = "loggerhead sea turtle" (33)

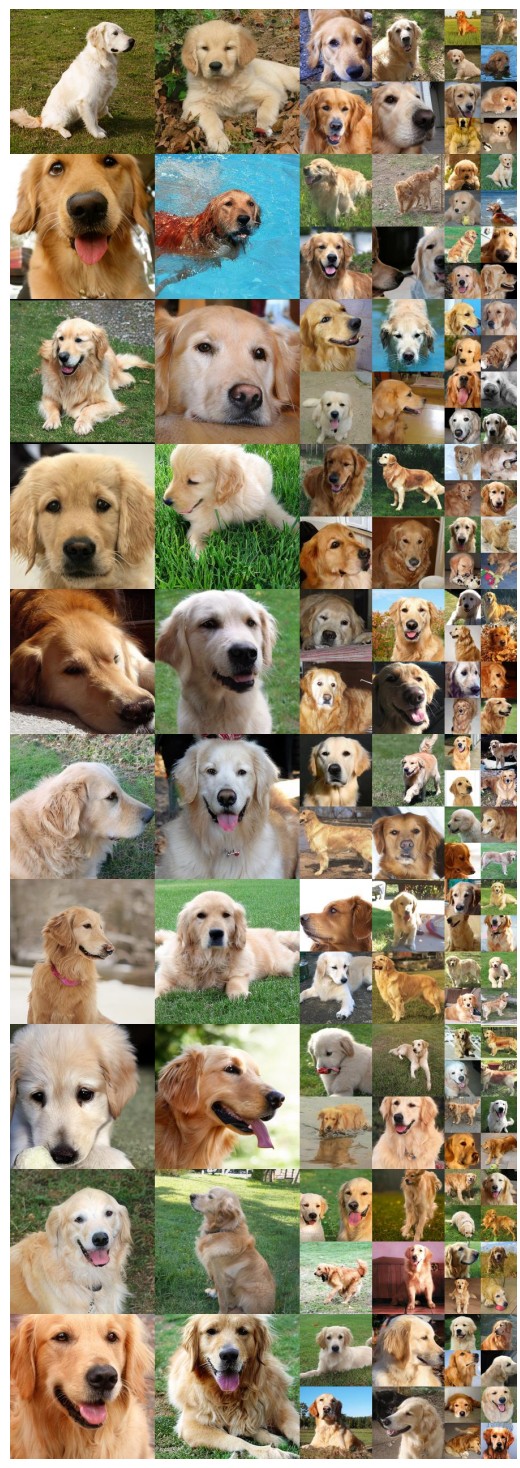 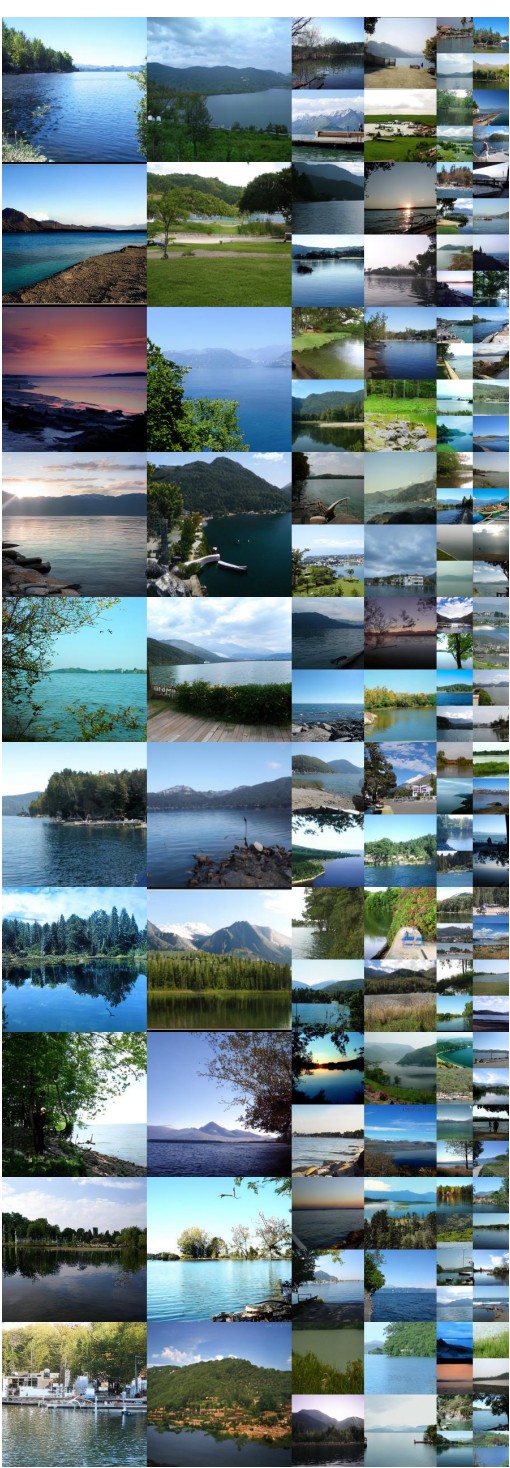

Figure 26: **Uncurated** $256 \times 256$ **DiCo-XL samples.**
Classifier-free guidance scale = 2.0
Class label = "golden retriever" (207)

Figure 27: **Uncurated** $256 \times 256$ **DiCo-XL samples.**
Classifier-free guidance scale = 2.0
Class label = "lake shore" (975)

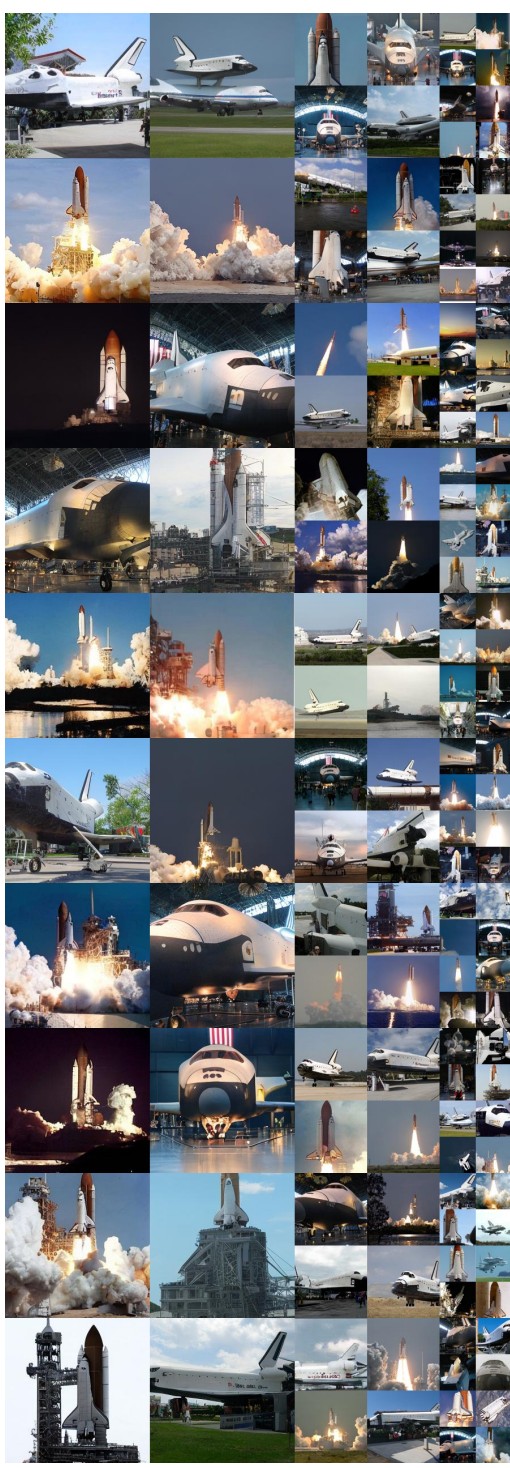

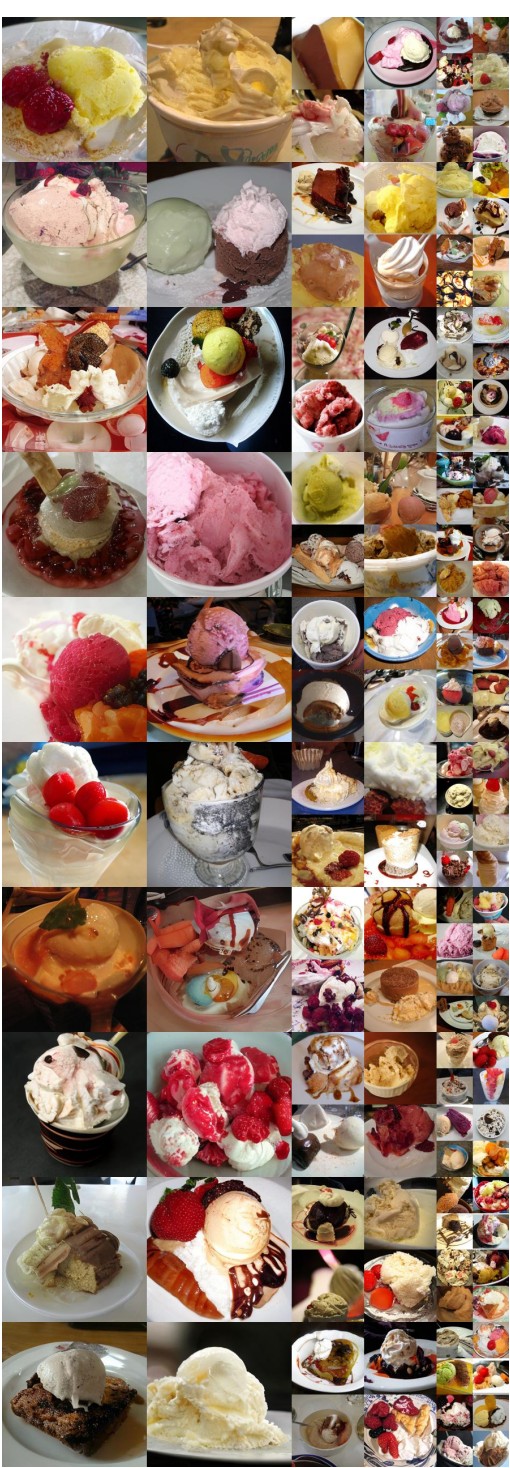

Figure 28: **Uncurated** $256 \times 256$ **DiCo-XL samples.**
Classifier-free guidance scale = 1.5
Class label = "space shuttle" (812)

Figure 29: **Uncurated** $256 \times 256$ **DiCo-XL samples.**
Classifier-free guidance scale = 1.5
Class label = "ice cream" (928)

