# OpenReview forum: "DiCo: Revitalizing ConvNets for Scalable and Efficient Diffusion Modeling"
_NeurIPS.cc/2025/Conference — NeurIPS 2025 spotlight_

### Official Review · Reviewer_c9AS · 2025-06-09

**Clarity:** 2
**Significance:** 4
**Originality:** 1
**Rating:** 5
**Confidence:** 4

**Summary:**

Diffusion Transformer (DiT), a promising diffusion model for visual generation, demonstrates impressive performance but incurs significant computational overhead. Intriguingly, analysis of pre-trained DiT models reveals that global self-attention is often redundant, predominantly capturing local patterns. This paper proposes a pure convolution-based diffusion model, deemed as DiCo, to explore the cnn-like diffusion variant.

**Questions:**

see weakness

**Ethical Concerns:**

["NO or VERY MINOR ethics concerns only"]

**Final Justification:**

The author's response has addressed all my concerns, and I agree to raise the final score to accept.

**Limitations:**

see weakness

**Quality:**

3

**Strengths And Weaknesses:**

Strengths
1. The comparison is fair and reasonable.
2. The writing is clear and easy to understand.
3. The acceleration of DiCo is significant and obvious.

Weaknesses
1. Lack of text-to-image results. As convolution is naturally incompatible with attention, I am doubtful about the multimodal performance, like text-to-image tasks.
2. The necessity of the long residuals. UNet often converges faster with an inferior final performance.  As pointed out by DiT[1] and SD3[2], ***the U-Net inductive bias is not crucial(see the introduction section in DiT), and the U-shape limits further scaling (see Fig.4 in SD3)***.  Thus, DiCo looks completely like a technological regression.
3. The claim of being state-of-the-art (SoTA) is unjustified, as DiCo's performance significantly lags behind modern diffusion transformers such as lightning-dit[3].
4. There is no comparison with FlowDCN[4]. Deformable convolution is one of the most powerful convolution variants. It is appropriate to provide some discussions and comparisons.

[1] Peebles W, Xie S. Scalable diffusion models with transformers[C]//Proceedings of the IEEE/CVF international conference on computer vision. 2023: 4195-4205.

[2] Esser P, Kulal S, Blattmann A, et al. Scaling rectified flow transformers for high-resolution image synthesis[C]//Forty-first international conference on machine learning. 2024.

[3]Yao J, Yang B, Wang X. Reconstruction vs. generation: Taming optimization dilemma in latent diffusion models[C]//Proceedings of the Computer Vision and Pattern Recognition Conference. 2025: 15703-15712.

[4]Wang S, Li Z, Song T, et al. FlowDCN: Exploring DCN-like Architectures for Fast Image Generation with Arbitrary Resolution[J]. arXiv preprint arXiv:2410.22655, 2024.

---

> ### Author Rebuttal · Authors · 2025-07-30
>
> Thank you for your feedback. We appreciate your recognition of the effectiveness of our method and the writing of our paper. Here are our responses to your comments.
>
> ---
>
> # Main Comments
>
> > **Q1: Lack of text-to-image results.**
>
> **A1:** Thank you for your insightful suggestions. Given the limited time during the rebuttal phase, **we follow U-ViT [1] and REPA [2] to conduct small-scale text-to-image generation experiments.** Specifically, we adopt the same experimental setup as [1,2]: training and evaluating models from scratch on the MS-COCO dataset, using CLIP as the text encoder with a token length of 77.
>
> We investigate two different approaches for **incorporating textual features into DiCo.**
> 1. The first uses the widely adopted **cross-attention** mechanism, integrated into the DiCo architecture to fuse text and visual features.
> 2. The second **transforms CLIP text embeddings into dynamic depthwise convolution (DWC) kernels.** We pad the text embeddings to a length of 81, feed them through a learnable MLP, and reshape the output into a 9×9 kernel. This kernel dynamically modulates DiCo's features via depthwise convolution. In this way, we can construct a **fully convolutional** text-to-image DiCo model **without relying on any self-attention or cross-attention operations.** We provide its PyTorch implementation below:
>
> ***Algorithm A:*** *PyTorch code of text conditional depthwise convolution*
> ```python
> import torch
> import torch.nn.functional as F
>
> def text_conditional_dwconv(x, context):
>     # x: (B, C, H, W) input feature maps
>     # context: (B, 77, C) CLIP text embeddings after an MLP
>     # output: (B, C, H, W) output after depthwise convolution
>     B, C, H, W = x.shape
>     context_pad = torch.cat([context, context[:, -1:].expand(-1, 4, -1)], dim=1)  # (B, 81, C)
>     kernels = context_pad.reshape(B, 9, 9, C).permute(0, 3, 1, 2).reshape(B * C, 1, 9, 9)
>     x_flat = x.view(1, B * C, H, W)
>     output = F.conv2d(x_flat, kernels, padding=4, groups=B * C).view(B, C, H, W)
>     return output
> ```
>
> Both feature fusion modules are inserted after the Conv Module within each DiCo block. Following U-ViT, we match the model size and use a CFG scale of 2.0. All FID results of other methods reported below are taken from REPA [2].
>
> ***Table:*** *Comparison on MS-COCO for text-to-image generation*
> | Method|Token Mixing Type|Params| FLOPs |Throughput↑|FID w/ CFG↓
> |-|:-:|:-:|:-:|:-:|:-:
> |U-ViT-S/2 (Deep)|Full Self-Attn|58M|21.3G |377 it/s| 5.48
> |MM-DiT| Full Self-Attn|512M|62.4G |135 it/s|5.30
> |DiCo-CrossAttn (ours)|Conv + CrossAttn |67M|14.9G |594 it/s|**4.87**
> |DiCo-DWC (ours)|Full Conv|69M|**12.9G**|**659 it/s**|4.93
>
> Our DiCo **outperforms prior methods in both efficiency and generation quality.** Notably, using text conditional DWC in place of cross-attention further improves throughput while maintaining competitive performance. This suggests that the **fully convolutional DiCo** has the potential to serve as the backbone for large-scale **text-to-image diffusion models**. We will include these results in the final version of our paper.
>
> ---
>
> > **Q2: The necessity of the long residuals. UNet often converges faster with an inferior final performance. The U-Net inductive bias is not crucial (see introduction in DiT), and the U-shape limits further scaling (see Fig.4 in SD3).**
>
> **A2:** Thank you for this insightful observation. We would like to clarify the following points:
> 1. While the inductive bias of U-Net may become less crucial when ample training resources are available, it remains valuable for **reducing training costs in resource-constrained scenarios.**
> 2. Our proposed method is **not limited to the U-Net structure**. It also achieves **strong performance within isotropic architectures.**
>
> Below, we provide a more detailed explanation of these points.
>
> ***For point 1:*** We acknowledge that recent works such as DiT and SD3 have argued that the U-Net inductive bias is less critical when models are trained at large scales with ample computational resources. However, **in many practical settings where compute is constrained,** the U-Net-style long residual connections remain valuable for **accelerating convergence and improving training efficiency.**
>
> As shown in the table below, DiCo with a U-Shape architecture **achieves better performance than DiT** while requiring **significantly fewer training iterations.** This highlights that the U-Shape structure enables **faster convergence and lower training cost,** which is especially important in resource-constrained settings.
>
> ***Table:*** *Comparison on ImageNet 256x256 for different iterations*
> |Method|Arch|Iterations|FID w/ CFG↓|IS↑
> |-|:-:|:-:|:-:|:-:
> |DiT-XL/2|Isotropic|7000K|2.27|278.2
> |DiCo-XL|U-Shape|1500K|2.25|270.4
> |DiCo-XL|U-Shape|3750K|2.05|282.2
>
>
> Furthermore, we respectfully clarify that Fig. 4 in SD3 does not claim U-Net inherently limits scalability (see the FID and CLIP score curves). In fact, the SD3 paper explicitly notes that "Vanilla DiT underperforms UViT" (see Sec. 5.2.3, page 9). A similar observation is made in DreamOmni [3] (see Fig. 3 and Sec. 4, page 6). In practice, several large-scale diffusion models continue to use long residuals successfully. For example, DreamOmni [3] (U-Shape) and Vidu [4] (Isotropic with skip connections) both adopt architectures with long residuals at scale.
>
> ***For point 2:*** Equally important, we emphasize that **the U-Shape design is not our core architectural contribution.** As shown in Table 3 of the paper, **DiCo consistently outperforms DiT** in both performance and efficiency **across various architectural settings** (isotropic, isotropic with skip connections, and U-shape). Moreover, we demonstrate that our method can be effectively applied to build **isotropic diffusion models**, achieving superior performance (FID of 1.33 on ImageNet 256x256) compared to modern DiTs (refer to **A3**). This provides strong evidence that our method **generalizes well beyond U-Net designs and is not inherently tied to long residuals.**
>
> We will include the above discussion in the final version of the paper.
>
> ---
>
> > **Q3: The claim of being SOTA is unjustified, as DiCo's performance lags behind modern DiTs such as LightningDiT.**
>
> **A3:** We appreciate your feedback regarding our SOTA claim. Given the rapid advancements in modern DiTs, we agree that the SOTA statement in our paper should be revised. We also recognize the importance of comparing our method with recent DiTs such as LightningDiT. We find that after aligning with their experimental settings, our method achieves **competitive performance along with a clear efficiency advantage.** The details are shown as follows.
>
> In our paper, we adopt **the original DiT experimental setup** (e.g., using SD-VAE, standard DDPM training, fixed hyperparameters) to ensure a fair and controlled comparison with DiT, DiG, and DiC. However, **LightningDiT differs from DiT's setup in several aspects**—for example, it replaces the SD-VAE with VA-VAE, adopts the Rectified Flow training objective, and modifies training hyperparameters, etc.
>
> To fairly assess DiCo’s competitiveness under this enhanced setup, we construct **LightningDiCo,** which **replaces LightningDiT’s self-attention with our Conv Module,** while keeping all other settings identical. This version uses an **isotropic backbone** (no U-Net structure), aligning directly with LightningDiT.
>
> ***Table:*** *Comparison with modern DiTs on ImageNet 256x256*
> | Method |Epochs|Params|FLOPs|Throughput↑|FID w/ CFG↓|IS↑
> |-|:-:|:-:|:-:|:-:|:-:|:-:
> |REPA |800|675M|118.7G|77 it/s|1.42|**305.7**
> |LightningDiT|800|675M|118.8G|73 it/s|1.35|295.3
> |LightningDiCo (ours)|800|679M|118.3G|**201 it/s**|**1.33**|300.2
>
> LightningDiCo achieves **an FID score of 1.33** and offers a **~2.8× throughput** gain over modern DiTs. This demonstrates that our convolutional backbone can **match or exceed the generative quality of modern DiTs while being significantly more efficient.** We will include this comparison in the final version of the paper and revise the SOTA claim.
>
> ---
>
> > **Q4: Missing comparison with FlowDCN.**
>
> **A4:** Thank you for pointing this out. We agree that deformable convolution is a strong and widely used convolutional variant, and FlowDCN represents an important baseline in this space.
>
> To fairly compare with FlowDCN, we **retain its original isotropic architecture and training setup,** and replace its **MultiScale DCN**  with our proposed **Conv Module.** The results are shown below:
>
> ***Table:*** *Comparison on ImageNet 256x256 (400K iterations, Small models)*
> |Method|Params|FLOPs|Throughput↑|FID w/o CFG↓|IS↑
> |-|:-:|:-:|:-:|:-:|:-:
> |SiT-S/2|32.9M|6.1G|1234 it/s|57.6|24.8
> |FlowDCN-S/2|30.3M|4.4G|1194 it/s|54.6|26.4
> |Ours-S/2|31.3M|4.6G|**2489 it/s**|**52.1**|**29.2**
>
> Our Conv Module achieves **better generative performance and ~2.1× higher throughput** compared to MultiScale DCN, despite being structurally simpler and easier to implement. We will include this comparison in the final version of the paper and add a discussion of FlowDCN in the related work.
>
> ---
> Thank you again for helping us strengthen our work. We hope that our response has addressed your concerns. We look forward to further communication with you!
>
> [1] All Are Worth Words: A ViT Backbone for Diffusion Models. In CVPR, 2023.
>
> [2] Representation Alignment for Generation: Training Diffusion Transformers Is Easier Than You Think. In ICLR, 2025.
>
> [3] DreamOmni: Unified Image Generation and Editing. In CVPR, 2025.
>
> [4] Vidu: a highly consistent, dynamic and skilled text-to-video generator with diffusion models. arXiv preprint arXiv:2405.04233.

---

> > ### Comment · Reviewer_c9AS · 2025-08-04
> >
> > The author's response has addressed all my concerns, and I agree to raise the final score to accept.

---

> > > ### Author Response · Authors · 2025-08-04
> > > **Thank you for your positive feedback**
> > >
> > > Dear Reviewer c9AS,
> > >
> > > Thank you very much for your positive feedback and for raising your score for our submission. We greatly appreciate your thoughtful comments and valuable suggestions, which have significantly improved the quality of our work. We will carefully incorporate your feedback as we revise our final paper.
> > >
> > > Best regards,
> > >
> > > The Authors of Submission 1112

---

### Official Review · Reviewer_LJ6t · 2025-06-17

**Clarity:** 4
**Significance:** 3
**Originality:** 3
**Rating:** 5
**Confidence:** 4

**Summary:**

This paper proposes a new backbone for diffusion models, Diffusion ConvNet (DiCo), as an alternative to DiT. It replaces self-attention with convolutions and proposes the CCA module to reduce channel redundancy. On the ImageNet benchmarks, DiCo surpasses state-of-the-art diffusion models while achieving significant efficiency gains.

**Questions:**

1. Regarding the CCA Module: Given the functional overlap between CCA and existing channel attention mechanisms (e.g., channel self-attention) in modulating channel features and the seemingly minor performance differences in the ablation study, it would be beneficial to elucidate the specific innovations within CCA that are critical for its effectiveness.

2. Addressing Long-Range Dependencies and Generalizability. Given ConvNets' inherent locality, it is unclear how DiCo effectively captures the crucial long-range dependencies essential for generating high-quality images, especially in complex scenarios such as high-resolution image synthesis, multi-object scenes, or intricate image editing tasks. It would benefit from a discussion on how DiCo addresses this challenge.

3. Questions about the ablation experiment. Table 4 consistently shows FID values below 54 across various configurations, whereas baseline models (DiT-S/2, DiC-S, DiG-S/2) in Table 1 report FID scores exceeding 58. This substantial performance improvement observed in DiCo warrants further explanation.

**Ethical Concerns:**

["NO or VERY MINOR ethics concerns only"]

**Final Justification:**

The authors have addressed my major concerns through experiments. Consequently, I recommend acceptance of this manuscript.
While the COCO experiments (Q1) demonstrate promising results, the integration of DiCo with more complex prompts and image editing tasks warrants further exploration in future work.
I encourage the authors to release the code and models publicly to accelerate progress in this area.

**Limitations:**

YES

**Quality:**

3

**Strengths And Weaknesses:**

- Strengths:
1. DiCo models demonstrate substantial computational efficiency gains compared with DiTs, concurrently achieving SOTA performance on ImageNet benchmarks.
2. This work identifies the channel redundancy issue when introducing convolutions to DiTs and addresses it by introducing CCA.  The proposed solution appears to be reasonable.
3. Comprehensive ablation studies are provided, detailing both architectural design and the contribution of individual components.

- Weaknesses:
The experimental evaluation is primarily restricted to class-conditional image generation. While the authors acknowledge limitations for not extending evaluations to other prominent generative tasks (e.g., text-to-image synthesis), this circumscribes a comprehensive assessment of DiCo's broader applicability and generalizability across the diverse landscape of diffusion model applications.

---

> ### Author Rebuttal · Authors · 2025-07-30
>
> Thank you for your feedback. We appreciate your recognition of the motivation and effectiveness of our method. Here are our responses to your comments.
>
> ---
>
> # Main Comments
>
> > **Q1: Lack of experiments for text-to-image generation.**
>
> **A1:** Thank you for your insightful suggestions. Given the limited time during the rebuttal phase, **we follow U-ViT [1] and REPA [2] to conduct small-scale text-to-image generation experiments.** Specifically, we adopt the same experimental setup as [1,2]: training and evaluating models from scratch on the MS-COCO dataset, using CLIP as the text encoder with a token length of 77.
>
>
> We investigate two different approaches for **incorporating textual features into DiCo.**
> 1. The first uses the widely adopted **cross-attention** mechanism, integrated into the DiCo architecture to fuse text and visual features.
> 2. The second **transforms CLIP text embeddings into dynamic depthwise convolution (DWC) kernels.** We pad the text embeddings to a length of 81, feed them through a learnable MLP, and reshape the output into a 9×9 kernel. This kernel dynamically modulates DiCo's features via depthwise convolution. In this way, we can construct a **fully convolutional** text-to-image DiCo model **without relying on any self-attention or cross-attention operations.** We provide its PyTorch implementation below:
>
> ***Algorithm:*** *PyTorch code of text conditional depthwise convolution*
> ```python
> import torch
> import torch.nn.functional as F
>
> def text_conditional_dwconv(x, context):
>     # x: (B, C, H, W) input feature maps
>     # context: (B, 77, C) CLIP text embeddings after an MLP
>     # output: (B, C, H, W) output after depthwise convolution
>     B, C, H, W = x.shape
>     context_pad = torch.cat([context, context[:, -1:].expand(-1, 4, -1)], dim=1)  # (B, 81, C)
>     kernels = context_pad.reshape(B, 9, 9, C).permute(0, 3, 1, 2).reshape(B * C, 1, 9, 9)
>     x_flat = x.view(1, B * C, H, W)
>     output = F.conv2d(x_flat, kernels, padding=4, groups=B * C).view(B, C, H, W)
>     return output
> ```
>
> Both feature fusion modules are inserted after the Conv Module within each DiCo block. Following U-ViT, we match the model size and use a CFG scale of 2.0. All FID results  of other methods reported below are taken from REPA [2].
>
> ***Table:*** *Comparison on MS-COCO for text-to-image generation*
> | Method|Token Mixing Type | Params | FLOPs | Throughput↑ | FID w/ CFG↓|
> |-|:-:|:-:|:-:|:-:|:-:|
> | U-ViT-S/2 (Deep) |  Full Self-Attn |58M   |   21.3G       | 377 it/s | 5.48 |
> | MM-DiT | Full Self-Attn| 512M  |62.4G |135 it/s | 5.30 |
> | DiCo-CrossAttn (ours)|Conv + CrossAttn | 67M   |  14.9G    | 594 it/s | **4.87** |
> | DiCo-DWC (ours) |Full Conv| 69M   |  **12.9G**    | **659 it/s** | 4.93 |
>
> Our DiCo **outperforms prior methods in both efficiency and generation quality.** Notably, using text conditional DWC in place of cross-attention further improves throughput while maintaining competitive performance. This suggests that the **fully convolutional DiCo** has the potential to serve as the backbone for large-scale **text-to-image diffusion models**. We will include these results in the final version of our paper.
>
> ---
>
> > **Q2: Functional overlap between CCA and existing channel attention mechanisms (e.g., channel self-attention).**
>
> **A2:** Thank you for raising this insightful point. We acknowledge that CCA and channel self-attention both aim to modulate channel-wise feature importance. However, our proposed **CCA is specifically designed to be more efficient,** particularly at large model scales, while retaining comparable generative performance.
>
> Unlike standard channel self-attention, which introduces quadratic complexity with respect to channel numbers, **CCA adopts a simplified, structured attention mechanism that avoids heavy pairwise computations.** This design choice enables substantial computational savings without sacrificing effectiveness. The comparison between CCA and channel self-attention at the XL model scale is shown below:
>
> ***Table:*** *CCA vs. channel self-attention on ImageNet 256x256 (400K iterations, XL models)*
> | Method | Params | FLOPs | Throughput↑ | FID w/o CFG↓  | IS↑  |
> |-|:-:|:-:|:-:|:-:|:-:|
> | Channel self-attention | 701M | 109.5G   | 150 it/s |11.71|98.76|
> | CCA  | 701M | **87.3G** | **208 it/s** | **11.67**| **100.42**|
>
> CCA achieves a **~1.4× speedup with slightly better FID and IS**, demonstrating its advantage as a lightweight and scalable alternative to conventional channel attention. We will include this analysis and additional clarification on CCA’s design motivation in the final version of the paper.
>
> ---
>
> > **Q3: Discussion of addressing long-range dependencies and generalizability.**
>
> **A3:**  Thank you for highlighting this important point. We acknowledge that the local receptive field of convolutional networks may limit their ability to capture long-range dependencies, especially in high-resolution settings.
>
> To address this, we explore two complementary solutions in DiCo:
> 1. **Enlarging convolutional kernel sizes:** A straightforward yet effective way to expand the receptive field is to use larger convolution kernels. As shown in Table 4 of our paper, increasing the kernel size leads to consistent improvements in generation quality, albeit with a trade-off in computational efficiency. This approach enables DiCo to capture broader context while maintaining the simplicity of convolution.
> 2. **Employing high-compression autoencoders:** An alternative solution is to leverage modern latent diffusion frameworks with **highly compressed latent spaces,** which reduces the burden on the backbone to model long-range spatial dependencies directly. Recent works such as VA-VAE [3] and DC-AE [4] propose autoencoders with downsampling ratios of up to 16× or 32×, which effectively shrink the spatial dimensions.
>
> We conduct experiments with **VA-VAE on ImageNet 256×256** and **DC-AE on ImageNet 512×512** following their experimental setups. The results are shown below:
>
> ***Table:*** *Comparison on ImageNet 256x256 using different Autoencoders*
> | Method    | Autoencoder | Downsample Ratio |Latent Size  | FID w/ CFG↓  | IS↑  |
> |-|:-:|:-:|:-:|:-:|:-:|
> | DiT-XL/2 | SD-VAE  |8x | 4x32x32   | 2.27   |  278.2 |
> | DiCo-XL (ours)| SD-VAE|8x | 4x32x32  |  2.05  | 282.2 |
> | DiCo-XL (ours)| VA-VAE|16x| 32x16x16|  **1.33**  | **300.2** |
>
> ***Table:*** *Comparison on ImageNet 512x512 using different Autoencoders*
> |Method| Autoencoder | Downsample Ratio |Latent Size|FID w/ CFG↓| IS↑  |
> |-|:-:|:-:|:-:|:-:|:-:|
> | DiT-XL/2 | SD-VAE  |8x  | 4x64x64 | 3.04 | 240.8|
> | DiCo-XL (ours)| SD-VAE|8x | 4x64x64  |2.53 | 275.7 |
> | DiCo-XL (ours)| DC-AE |32x |32x16x16|**1.78**| **280.1**|
>
> The use of high-compression autoencoders mitigates the locality limitation of convolution and enables DiCo to model long-range dependencies more effectively. Notably, DiCo achieves better **FID scores of 1.33 on ImageNet 256×256 and 1.78 on ImageNet 512×512** with these autoencoders. We hope to further explore the application of DiCo in more complex scenarios in future work. We will include the discussion in the final version of our paper.
>
> ---
>
> > **Q4: Table 4 consistently shows FID values below 54 across various configurations, whereas baseline models (DiT-S/2, DiC-S, DiG-S/2) in Table 1 report FID scores exceeding 58.**
>
> **A4:** Thank you for your careful review. The difference in FID values between Table 1 and Table 4 primarily stems from **architecture settings.** In Table 1, both DiCo and DiC are based on the U-Shape architecture, while DiT and DiG adopt isotropic architectures. In the ablation studies presented in Table 4, we keep the U-Shape architecture fixed while ablating different modules within DiCo. Moreover, in Table 3, we show that DiCo consistently achieves superior performance across different architectural settings. To make this distinction more explicit, **we summarize the fair comparisons below by isolating the effect of architecture:**
>
> ***Table:*** *Comparison on ImageNet 256x256 (Isotropic, 400K iterations, Small models)*
> | Method   |  Arch  | Params | FLOPs | Throughput↑ |FID w/o CFG↓| IS↑|
> |-|:-:|:-:|:-:|:-:|:-:|:-:|
> | DiT (Self-Attn) | Isotropic |32.9M    | 6.1G  |  1087 it/s | 67.2 |  20.4 |
> | DiG (Gated Linear Attn) |  Isotropic|33.0M | 4.3G | 961 it/s| 62.1 |  22.8 |
> | DiCo (Conv Module)|Isotropic| 33.7M| 5.7G| **1901 it/s** | **60.6**  | **25.4** |
>
> ***Table:*** *Comparison on ImageNet 256x256 (U-Shape, 400K iterations, Small models)*
> | Method|Arch|Params|FLOPs|Throughput↑|FID w/o CFG↓|IS↑|
> |-|:-:|:-:|:-:|:-:|:-:|:-:|
> | DiT (Self-Attn) | U-Shape |33.0M    | 4.2G  |  1141 it/s | 54.0 |28.5|
> | DiG (Gated Linear Attn) | U-Shape |33.1M    | 4.2G  |937 it/s| 53.2| 28.9  |
> | Window Attn | U-Shape |33.1M| 4.2G|1165 it/s| 53.2|28.3|
> | Focused Linear Attn | U-Shape |32.9M| 4.3G  |971 it/s| 50.6|30.8|
> |Agent Attn| U-Shape | 33.3M| 4.3G| 1161 it/s | 52.1 |28.8|
> | DiCo (Conv Module, w/o CCA)|U-Shape|33.0M|4.2G|**1731 it/s**|54.8|28.4|
> | DiCo (Conv Module)|U-Shape|33.0M|4.2G|1696 it/s|**50.0**|**31.4**|
>
> **Under a fair comparison with identical architectures, our DiCo consistently outperforms the baseline methods.** This indicates that the **performance and efficiency gains of DiCo stem from our proposed Conv Module and CCA.** We will clarify this point more explicitly in the final version of the paper.
>
> ---
>
> Thank you again for helping us strengthen our work. We hope that our response has addressed your concerns. We look forward to further communication with you!
>
> [1] All Are Worth Words: A ViT Backbone for Diffusion Models. In CVPR, 2023.
>
> [2] Representation Alignment for Generation: Training Diffusion Transformers Is Easier Than You Think. In ICLR, 2025.
>
> [3] Reconstruction vs. Generation: Taming Optimization Dilemma in Latent Diffusion Models. In CVPR, 2025.
>
> [4] Deep Compression Autoencoder for Efficient High-Resolution Diffusion Models. In ICLR, 2025.

---

> > ### Comment · Reviewer_LJ6t · 2025-08-07
> >
> > This response addresses my concerns and I raise my score.

---

> > > ### Author Response · Authors · 2025-08-07
> > > **Thank you for your positive feedback**
> > >
> > > Dear Reviewer LJ6t,
> > >
> > > Thank you very much for your positive feedback and for raising your score for our submission. We sincerely appreciate your thoughtful comments and are delighted that our rebuttal has addressed your concerns. We will carefully consider your suggestions and incorporate them as we revise the final version of our paper.
> > >
> > > Best regards,
> > >
> > > The Authors of Submission 1112

---

### Official Review · Reviewer_yq4n · 2025-07-03

**Clarity:** 3
**Significance:** 4
**Originality:** 4
**Rating:** 6
**Confidence:** 3

**Summary:**

In this paper,, the authors observe that self-attention in diffusion transformers seems to mostly capture local patterns, motivating going back to a convolutional architecture that could be more efficient. However, naively replacing the self attention layers with convolutional layers degrades performance, and the authors observe many channels remaining inactive during generation. Therefore, the authors introduce a compact channel attention (CCA) mechanism, which dynamically activates informative channels with lightweight linear projection. This enables this convolutional architecture achieve SOTA results on variety of benchmarks.

**Questions:**

My understanding is all the experiments shown here are for latent diffusion. do the observations about local attention patterns and improvements with CCA also hold true for pixel-level diffusion?

are the baselines in table 1 with CFG or not?

a big advantage of using a convolutional arch is that the scaling of computation with resolution will not be quadratic like with self-attention. Could a comparison at a larger scale like 1024x1024 be done to demonstrate this?

**Ethical Concerns:**

["NO or VERY MINOR ethics concerns only"]

**Final Justification:**

addressed my suggestions

**Limitations:**

yes

**Quality:**

4

**Strengths And Weaknesses:**

Strengths:
* well-structured and well-motivated
* compelling results on benchmarks
* unique architectural modifications

Weaknesses:
* no text-to-image results, even small-scale
* only 256x256 and 512x512 results on ImageNet

---

> ### Author Rebuttal · Authors · 2025-07-30
>
> Thank you for your feedback. We appreciate your recognition of our method's innovation and effectiveness. Here are our responses to your comments.
>
> ---
>
> # Main Comments
>
> > **Q1: No text-to-image generation results, even small-scale.**
>
> **A1:** Thank you for your insightful suggestions. Given the limited time during the rebuttal phase, **we follow U-ViT [1] and REPA [2] to conduct small-scale text-to-image generation experiments.** Specifically, we adopt the same experimental setup as [1,2]: training and evaluating models from scratch on the MS-COCO dataset, using CLIP as the text encoder with a token length of 77.
>
> We investigate two different approaches for **incorporating textual features into DiCo.**
> 1. The first uses the widely adopted **cross-attention** mechanism, integrated into the DiCo architecture to fuse text and visual features.
> 2. The second **transforms CLIP text embeddings into dynamic depthwise convolution (DWC) kernels.** We pad the text embeddings to a length of 81, feed them through a learnable MLP, and reshape the output into a 9×9 kernel. This kernel dynamically modulates DiCo's features via depthwise convolution. In this way, we can construct a **fully convolutional** text-to-image DiCo model **without relying on any self-attention or cross-attention operations.** We provide its PyTorch implementation below:
>
>
> ***Algorithm:*** *PyTorch code of text conditional depthwise convolution*
> ```python
> import torch
> import torch.nn.functional as F
>
> def text_conditional_dwconv(x, context):
>     # x: (B, C, H, W) input feature maps
>     # context: (B, 77, C) CLIP text embeddings after an MLP
>     # output: (B, C, H, W) output after depthwise convolution
>     B, C, H, W = x.shape
>     context_pad = torch.cat([context, context[:, -1:].expand(-1, 4, -1)], dim=1)  # (B, 81, C)
>     kernels = context_pad.reshape(B, 9, 9, C).permute(0, 3, 1, 2).reshape(B * C, 1, 9, 9)
>     x_flat = x.view(1, B * C, H, W)
>     output = F.conv2d(x_flat, kernels, padding=4, groups=B * C).view(B, C, H, W)
>     return output
> ```
>
> Both feature fusion modules are inserted after the Conv Module within each DiCo block. Following U-ViT, we match the model size and use a CFG scale of 2.0. All FID results  of other methods reported below are taken from REPA [2].
>
> ***Table:*** *Comparison on MS-COCO for text-to-image generation*
> | Method|Token Mixing Type | Params | FLOPs | Throughput↑ | FID w/ CFG↓|
> |-|:-:|:-:|:-:|:-:|:-:|
> | U-ViT-S/2 (Deep)|  Full Self-Attn |58M   |   21.3G       | 377 it/s | 5.48 |
> | MM-DiT | Full Self-Attn| 512M  |62.4G |135 it/s | 5.30 |
> | DiCo-CrossAttn (ours)|Conv + CrossAttn | 67M   |  14.9G    | 594 it/s | **4.87** |
> | DiCo-DWC (ours) |Full Conv| 69M   |  **12.9G**    | **659 it/s** | 4.93 |
>
> Our DiCo **outperforms prior methods in both efficiency and generation quality.** Notably, using text conditional DWC in place of cross-attention further improves throughput while maintaining competitive performance. This suggests that the **fully convolutional DiCo** has the potential to serve as the backbone for large-scale **text-to-image diffusion models**. We will include these results in the final version of our paper.
>
> ---
>
> > **Q2: Applicability to pixel-level diffusion.**
>
> **A2:** Thank you for your thoughtful question. Yes, the effectiveness of our method also holds in the pixel-level diffusion setting.
>
> Following U-ViT’s setup, we conduct experiments on the ImageNet 64×64 dataset using pixel-level diffusion. We adjust DiCo’s width and depth to match the overall model size of U-ViT-L/4 for a fair comparison. As shown below, DiCo achieves significantly better performance and efficiency:
>
> ***Table:*** *Pixel-level diffusion on ImageNet 64×64*
> | Method      | Params | FLOPs | Throughput↑ | FID w/o CFG↓  | IS↑  |
> |------------|:--------:|:--------:|:------------:|:------:|:------:|
> | U-ViT-L/4  | 287M    | 76.5G   | 117 it/s | 4.26 |  40.66 |
> | DiCo  | 281M    | **75.1G**   | **201 it/s** | **3.07**  | **48.32** |
>
> These results confirm that DiCo generalizes well to pixel-level diffusion. The benefits of convolutional locality and lightweight channel-wise modulation (via CCA) remain consistent in this setting. We will include these results in the final version of the paper.
>
> ---
>
> > **Q3: Are the baselines in Table 1 with CFG or not?**
>
> **A3:** Thank you for the careful observation. In Table 1, **all models except those trained for 400K iterations use CFG during evaluation.** The **400K-iteration models are intended for early-stage comparison and are reported without CFG**, consistent with prior works (e.g., DiT, DiG).
>
> All baseline metrics are taken directly from their respective published papers to ensure fairness. We will clarify the use of CFG in Table 1 and the main text in the final version of the paper.
>
> ---
>
> > **Q4: Comparison with DiT at a larger resolution like 1024x1024.**
>
> **A4:** Thank you for the valuable suggestion. We conduct experiments on ImageNet 1024×1024. Since the original DiT paper does not report results at this resolution, we train both DiT and DiCo from scratch under the same settings for a fair comparison. Due to DiT’s significantly slower training speed at high resolutions, we use small-scale models and train them for 400K iterations. The results are shown below:
>
> ***Table:*** *Comparison on ImageNet 1024x1024 (400K iterations, Small models)*
> | Method      | Params | FLOPs | Throughput↑ | FID w/o CFG↓  | IS↑  |
> |------------|:--------:|:--------:|:------------:|:------:|:------:|
> | DiT-S/2 | 33M    | 241.8G   | 28 it/s | 105.82 |  12.59 |
> | DiCo-S  | 33M    | **67.8G**   | **143 it/s** | **95.64**  | **16.44** |
>
> These results clearly show that DiCo scales much more efficiently to high resolutions—achieving a **5× speedup** with better generation quality. We will include these findings in the final version of the paper.
>
> ---
>
> Thank you again for helping us strengthen our work. We hope that our response has addressed your concerns. We look forward to further communication with you!
>
> [1] All Are Worth Words: A ViT Backbone for Diffusion Models. In CVPR, 2023.
>
> [2] Representation Alignment for Generation: Training Diffusion Transformers Is Easier Than You Think. In ICLR, 2025.

---

> > ### Comment · Area_Chair_MZ6X · 2025-08-05
> >
> > Dear Reviewer yq4n,
> >
> > Please take a moment to carefully read the authors’ rebuttal.
> > If the rebuttal addresses your concerns as expected, kindly reply and consider updating your score accordingly.
> > If not, please indicate what additional clarification or information you would need from the authors.
> >
> > Thank you for your time and contribution.

---

> > ### Comment · Reviewer_yq4n · 2025-08-06
> >
> > This response addresses my concerns and I raise my score to strong accept.

---

> > > ### Author Response · Authors · 2025-08-06
> > > **Thank you for your positive feedback**
> > >
> > > Dear Reviewer yq4n,
> > >
> > > Thank you very much for your positive feedback and for increasing your score for our submission. We greatly appreciate your thoughtful comments and are pleased that our rebuttal addressed your concerns. We will carefully incorporate your suggestions as we revise our final paper.
> > >
> > > Best regards,
> > >
> > > The Authors of Submission 1112

---

### Official Review · Reviewer_3NNh · 2025-07-04

**Clarity:** 3
**Significance:** 2
**Originality:** 2
**Rating:** 4
**Confidence:** 4

**Summary:**

This paper replace self-attention layer in DiT with a novel CNN block which is enhanced by proposed compact channel attention (CAA) mechanism. By this means, the proposed model DiCo outperforms DiT on higher performance and efficiency at class-conditional image generation task.

**Questions:**

1. Insufficient innovation in the model architecture. The proposed DiCo seems to be the fusion of Stable Diffusion U-shape architecture and CNN-version DiT block. The highlighed CCA mechanism is quite similar to SE block. The biggest difference is removing the channel compression FC layer? Although combining different methods to achieve new functions can be considered as an innovation, it is undoubtedly that the degree of this mode of innovation is not sufficient enough for NeurIPS.
2. Lacking comparison with U-net based diffusion models.  Since DiCo owns the U-shape architecture and the DiCo blocks are actually based on convolution module, why not conduct comparison with early U-net based diffusion, such as classic Stable Diffusion v1.5 and xl? Since authors have compared DiCo with DiT, why not compare to DiT based model such as Stable Diffusion 3 and FLUX? Though these models are original for text-to-image generation, I think simply replacing text prompt encoder with class embedder of DiT can enable the fair comparison.
3. Inappropriate abbreviation. Authors abbreviated convolutional neural networks as ConvNets, which easily lead readers to mistake it for ConvNeXt. I believe it is a commen sense in the deep learning community that the abbreviation of convolutional neural networks is "CNN". If the authors insist on using the abbreviation "ConvNets", please provide a convincing reason for it.

**Ethical Concerns:**

["NO or VERY MINOR ethics concerns only"]

**Final Justification:**

4: Borderline accept

**Limitations:**

yes

**Quality:**

3

**Strengths And Weaknesses:**

Strengths:
1. Considerable implemention details and experimental setting;
2. Reader-friendly good  and clear writing;
3. Visual insights, especially finding of localized representations in Figure4.

Weakness:
1. Insufficient innovation in the model architecture.
2. Lacking comparison with U-net based diffusion models.
3. Inappropriate abbreviation.

---

> ### Author Rebuttal · Authors · 2025-07-30
>
> Thank you for your feedback. We appreciate your recognition of the value of visual insights and the adequacy of experiments. Here are our responses to your comments.
>
> ---
>
> # Main Comments
>
> > **Q1: Insufficient innovation in the model architecture.**
>
> **A1:** Thank you for your valuable comments. From a broader perspective, our work **challenges a popular assumption** in today’s community: that Vision Transformers (e.g., DiT) are inherently superior to convolutional architectures for generative modeling. Our proposed DiCo demonstrates that a **purely convolutional design** can not only match but **outperform ViT-based diffusion models** like DiT in terms of both quality and efficiency. This offers a **fresh and timely perspective** to the research community, which in itself constitutes an important form of innovation.
>
> Moreover, we should know that **the technical innovation should be consistent with the research background and motivation.** In terms of model design, our innovations are reflected in both the **overall architecture** and **key components:**
>
> 1. **Overall architecture:** DiT is a pioneering diffusion model that applies the ViT as its backbone. Building on this line of research, DiCo introduces a **fully convolutional backbone** for diffusion models, inspired by the **localized nature of attention** in DiT. This design remains largely unexplored in the diffusion-based generative modeling literature. Furthermore, we extend DiCo to the text-to-image generation setting (refer to **A2**), resulting in **a fully convolutional text-to-image model.** This is a significant departure from existing text-to-image architectures, which predominantly rely on different attention mechanisms.
> 2. **Component-level design:** We identify a specific challenge in convolutional designs—**channel redundancy**—and propose a simple yet effective solution: the CCA module. While CCA shares some similarities with SE modules, it removes the FC layers and RELU non-linearities, resulting in a lighter and more efficient design. As shown in Table 4, CCA improves both generation quality and throughput compared to SE, validating its suitability in generative settings.
>
>
> Overall, our model offers fresh insights to the research community and introduces a novel design perspective, through which it showcases meaningful technical innovation.
>
> ---
>
> > **Q2: Lacking comparison with Stable Diffusion's UNet and FLUX's MM-DiT.**
>
> **A2:** Thank you for raising this important point. We agree that comparing DiCo with Stable Diffusion's UNet and FLUX's MM-DiT is essential for evaluating its effectiveness.
>
> First, regarding FLUX's MM-DiT: its architecture is a variant of DiT, with the primary difference lying in how textual features are handled. When replacing the text prompt encoder with a class embedding module (as in class-conditional ImageNet generation), the architecture effectively reduces to DiT. Therefore, **we consider MM-DiT a meaningful representative of DiT-style backbones in the text-to-image setting.**
>
> Second, we acknowledge the importance of comparing against strong baselines such as Stable Diffusion's UNet and FLUX's MM-DiT. Given the limited time during the rebuttal phase, we follow prior works (U-ViT [1], REPA [2]) and **conduct a fair small-scale text-to-image evaluation using Stable Diffusion’s U-Net and FLUX's MM-DiT as baselines.** Specifically, we adopt the same experimental setup as [1,2]: training and evaluating models from scratch on the MS-COCO dataset, using CLIP as the text encoder with a token length of 77.
>
> We investigate two different approaches for **incorporating textual features into DiCo.**
> 1. The first uses the widely adopted **cross-attention** mechanism, integrated into the DiCo architecture to fuse text and visual features.
> 2. The second **transforms CLIP text embeddings into dynamic depthwise convolution (DWC) kernels.** We pad the text embeddings to a length of 81, feed them through a learnable MLP, and reshape the output into a 9×9 kernel. This kernel dynamically modulates DiCo's features via depthwise convolution. In this way, we can construct a **fully convolutional** text-to-image DiCo model **without relying on any self-attention or cross-attention operations.** We provide its PyTorch implementation below:
>
> ***Algorithm:*** *PyTorch code of text conditional depthwise convolution*
> ```python
> import torch
> import torch.nn.functional as F
>
> def text_conditional_dwconv(x, context):
>     # x: (B, C, H, W) input feature maps
>     # context: (B, 77, C) CLIP text embeddings after an MLP
>     # output: (B, C, H, W) output after depthwise convolution
>     B, C, H, W = x.shape
>     context_pad = torch.cat([context, context[:, -1:].expand(-1, 4, -1)], dim=1)  # (B, 81, C)
>     kernels = context_pad.reshape(B, 9, 9, C).permute(0, 3, 1, 2).reshape(B * C, 1, 9, 9)
>     x_flat = x.view(1, B * C, H, W)
>     output = F.conv2d(x_flat, kernels, padding=4, groups=B * C).view(B, C, H, W)
>     return output
> ```
>
> Both fusion modules are inserted after the Conv Module in each DiCo block. Following U-ViT, we match model size and use a CFG scale of 2.0. The FID result for Stable Diffusion’s U-Net is taken from U-ViT [1], and that for FLUX's MM-DiT is from REPA [2].
>
> ***Table:*** *Comparison on MS-COCO for text-to-image generation*
> | Method|Token Mixing Type | Params | FLOPs | Throughput↑ | FID w/ CFG↓|
> |-|:-:|:-:|:-:|:-:|:-:|
> | U-ViT-S/2 (Deep) |  Full Self-Attn |58M   |   21.3G       | 377 it/s | 5.48 |
> | Stable Diffusion's UNet |  Conv + Self-Attn + CrossAttn |53M   |   20.9G       | 441 it/s | 7.32 |
> | FLUX's MM-DiT | Full Self-Attn| 512M  |62.4G |135 it/s | 5.30 |
> | DiCo-CrossAttn (ours)|Conv + CrossAttn | 67M   |  14.9G    | 594 it/s | **4.87** |
> | DiCo-DWC (ours) |Full Conv| 69M   |  **12.9G**    | **659 it/s** | 4.93 |
>
> DiCo **outperforms both Stable Diffusion's UNet and FLUX's MM-DiT** in terms of efficiency and generation quality. Notably, using text conditional DWC in place of cross-attention further improves throughput while maintaining competitive performance. This suggests that the **fully convolutional DiCo** has the potential to serve as the backbone for large-scale **text-to-image diffusion models**. We will include these results in the final version of our paper.
>
> ---
>
> > **Q3: Inappropriate abbreviation.**
>
> **A3:** Thank you for pointing this out. We understand the concern that the abbreviation "ConvNet" could be confused with "ConvNeXt," and we agree that "CNN" is the more widely recognized and unambiguous term in the deep learning community. While “ConvNet” has been used in several prior works (e.g., ConvNeXt [3], RepVGG [4]), we acknowledge that using "CNN" would provide greater clarity for readers. We will revise the manuscript to replace "ConvNet" with "CNN" throughout, in order to avoid confusion and maintain consistency with common terminology.
>
> ---
> Thank you again for helping us strengthen our work. We hope that our response has addressed your concerns. We look forward to further communication with you!
>
> [1] All Are Worth Words: A ViT Backbone for Diffusion Models. In CVPR, 2023.
>
> [2] Representation Alignment for Generation: Training Diffusion Transformers Is Easier Than You Think. In ICLR, 2025.
>
> [3] A ConvNet for the 2020s. In CVPR, 2022.
>
> [4] RepVGG: Making VGG-style ConvNets Great Again. In CVPR, 2021.

---

> > ### Comment · Reviewer_3NNh · 2025-08-04
> >
> > The author's reply solved my problem.

---

> > > ### Author Response · Authors · 2025-08-04
> > > **Thank you for your positive feedback**
> > >
> > > Dear Reviewer 3NNh,
> > >
> > > We sincerely appreciate your thoughtful comments and valuable suggestions. We are grateful that our rebuttal was able to address your concerns. We will carefully incorporate your comments as we revise our final paper.
> > >
> > > Best regards,
> > >
> > > The Authors of Submission 1112

---

### Note · Authors · 2025-08-15

Dear Reviewers and Area Chairs,

After thorough discussions with all reviewers, we would like to express our sincere gratitude for the constructive engagement and thoughtful feedback, which have been invaluable in strengthening our work. We are pleased that **the rebuttal has addressed all reviewers’ concerns.**

> **Review Highlights**

We appreciate that the reviewers recognized and valued the key strengths of our work:

- **Novel Insight:** We demonstrate that a fully convolutional design can surpass ViT-based diffusion models in both quality and efficiency, challenging the popular assumption that ViTs are inherently superior for generative modeling.
- **Clear Motivation:** Through analysis of pre-trained DiT models, we uncover significant redundancy and locality in their global attention, motivating the development of more efficient architectures.
- **Unique Solution:** We identify the channel redundancy issue arising from introducing convolutions to DiTs and address it using a compact channel attention mechanism.
- **Superior Performance:** Compared to various transformer-based diffusion models, our method achieves competitive generative quality while delivering substantial efficiency gains.

> **Main Concerns Addressed**

During the rebuttal phase, we provided further clarifications to address the major concerns:

- **Text-to-Image Results:** We explored text-to-image generation on MS-COCO and compared with SOTA methods (e.g., Stable Diffusion's UNet, FLUX's MM-DiT), further validating the effectiveness of our method in text-to-image settings.
- **High-Resolution Results:** We extended the evaluation to ImageNet 1024×1024, where DiCo demonstrates pronounced efficiency advantages at high resolutions.
- **Necessity of Long Residuals:** We examined the applicability of our method beyond U-Net architectures, showing that its effectiveness is not inherently dependent on long residuals.
- **Comparison with Modern DiTs:** We included comparisons with recent DiTs (LightningDiT and REPA), showing that DiCo matches their generative quality while significantly improving efficiency.


We sincerely thank the reviewers and area chairs for the time and effort in evaluating our work and for facilitating a constructive exchange during the discussion period. We believe the revisions and new results have substantially strengthened our submission and that our work will serve as a valuable contribution to the community.

Best regards,

The Authors of Submission 1112

---

### Decision · Program_Chairs · 2025-09-17

**Decision:**

Accept (spotlight)

**Comment:**

***(a) Scientific Claims and Findings***

This paper introduces DiCo, a family of convolutional diffusion models enhanced with a lightweight Compact Channel Attention (CCA) mechanism to address channel redundancy. The work challenges the prevailing assumption that Vision Transformers (ViTs) are inherently superior for diffusion modeling, showing that a fully convolutional design can match or surpass DiT-based models in both quality and efficiency. Empirical results demonstrate strong improvements in throughput and FID across ImageNet (256, 512, 1024 resolutions), text-to-image generation on MS-COCO, and pixel-level diffusion.

***(b) Strengths and Weaknesses***

The submission is well-written, clearly motivated, and supported by extensive ablations and comparisons. Strengths include the novel perspective on convolutional backbones, convincing efficiency advantages (especially at high resolutions), and thorough rebuttal clarifications. Weaknesses raised by reviewers included questions of novelty, lack of multimodal results, generalizability beyond ImageNet, and overstated SOTA claims.

***(c) Discussion and Rebuttal Summary***
+ Reviewer 3NNh: Concerned about novelty, lack of UNet comparisons, and terminology. Authors clarified innovation, added UNet/MM-DiT comparisons, and revised terminology. The reviewer was satisfied and supported acceptance.
+ Reviewer yq4n: Requested text-to-image and higher-resolution results. Authors added MS-COCO, pixel-level ImageNet 64×64, and ImageNet 1024×1024 experiments, addressing concerns. The reviewer raised the score to strong accept.
+ Reviewer LJ6t: Questioned CCA’s originality and DiCo’s generalizability. Authors showed efficiency advantages, discussed strategies for long-range dependencies, and clarified ablation settings. The reviewer raised the score to accept.
+ Reviewer c9AS: Flagged missing multimodal evaluation, U-Net reliance, SOTA claims, and FlowDCN comparisons. Authors added text-to-image experiments, clarified independence from U-Nets, compared against LightningDiT and FlowDCN, and softened SOTA claims. The reviewer raised the score to accept.

After the rebuttal, all reviewers increased their scores, resulting in 1 borderline accept, 2 accepts, and 1 strong accept.

***(d) Reasons to Final Recommendation: Accept (Spotlight)***

This work stands out for (1) challenging a dominant paradigm by reviving ConvNets for diffusion models, (2) providing strong empirical validation across diverse tasks, (3) demonstrating scalability to billion-parameter and multimodal settings, and (4) showing compelling efficiency gains at high resolutions (e.g., 5× faster at 1024×1024). These qualities make it a timely and influential contribution likely to inspire further research.